# The characteristics and structure of extra-tropical cyclones in a warmer climate

Victoria A. Sinclair[1], Mika Rantanen[1], Päivi Haapanala[1], Jouni Räisänen[1], and Heikki Järvinen[1]

[1]Institute for Atmospheric and Earth System Research / Physics, Faculty of Science, University of Helsinki, PO BOX 64, FI-00014

**Correspondence:** Victoria Sinclair (Victoria.Sinclair@helsinki.fi)

**Abstract.** Little is known about how the structure of extra-tropical cyclones will change in the future. In this study aquaplanet simulations are performed with a full complexity atmospheric model. These experiments can be considered as an intermediate step towards increasing knowledge of how, and why, extra-tropical cyclones respond to warming. A control simulation and a warm simulation in which the sea surface temperatures are increased uniformly by 4 K are run for 11 years. Extra-tropical cyclones are tracked, cyclone composites created, and the omega equation applied to assess causes of changes in vertical motion. Warming leads to a 3.3% decrease in the number of extra-tropical cyclones, no change to the median intensity nor life time of extra-tropical cyclones, but to a broadening of the intensity distribution resulting in both more stronger and more weaker storms. Composites of the strongest extra-tropical cyclones show that total column water vapour increases everywhere relative to the cyclone centre and that precipitation increases by up to 50% with the 4 K warming. The spatial structure of the composite cyclone changes with warming: the 900–700-hPa layer averaged potential vorticity, 700-hPa ascent and precipitation maximums associated with the warm front all move polewards and downstream and the area of ascent expands in the downstream direction. Increases in ascent forced by diabatic heating and thermal advection are responsible for the displacement whereas increases in ascent due to vorticity advection lead to the downstream expansion. Finally, maximum values of ascent due to vorticity advection and thermal advection weaken slightly with warming whereas those attributed to diabatic heating increase. Thus, cyclones in warmer climates are more diabatically driven.

## 1 Introduction

Extra-tropical cyclones (also referred to as mid-latitude cyclones) are a fundamental part of the atmospheric circulation in the mid-latitudes due to their ability to transport large amounts of heat, moisture and momentum. Climatologically, extra-tropical cyclones are responsible for most of the precipitation in the mid-latitudes with over 70% of precipitation in large parts of Europe and North America due to the passage of an extra-tropical cyclone (Hawcroft et al., 2012). Extra-tropical cyclones are also the primary cause for mid-latitude weather variability and can lead to strong winds. For example, a severe extra-tropical cyclone Kyrill moved over large parts of Northern Europe in 2007, bringing strong winds that results in 43 deaths and 6.7bn US dollars of insured damages (Fink et al., 2009). Intense extra-tropical cyclones can also be associated with heavy rain or snow which can result in floods and travel disruption. Thus, given the large social and economic impacts that extra-tropical

cyclones can cause, there is considerable research devoted to understanding the climatology and governing dynamics of these systems.

Many studies have investigated the spatial distribution and frequency of extra-tropical cyclones in the current climate by analysing reanalysis data sets (e.g. Simmonds and Keay, 2000; Hoskins and Hodges, 2002; Wernli and Schwierz, 2006) and consequently the location of the climatological mean storm tracks, in both hemispheres, in the current climate is well known. Climatologies of cyclone number and intensity in the current climate have also been created based on reanalysis data sets and numerous different objective cyclone tracking algorithms (Neu et al., 2013). Globally there is good agreement between methods for inter-annual variability of cyclone numbers and the shape of the cyclone intensity distribution but less agreement in terms of the total cyclone numbers, particularly in terms of weak cyclones.

The spatial structure of extra-tropical cyclones in the current climate has also been extensively examined (see Schultz et al. (2019) for an overview). The starting point was the development of the Norwegian cyclone model (Bjerknes, 1919), a conceptual framework describing the spatial and temporal evolution of an extra-tropical cyclone. As considerable variability was noted in cyclone structures, Shapiro and Keyser (1990) subsequently developed a sister conceptual model. In a different theme, Harrold (1973), Browning et al. (1973), and Carlson (1980) studied three-dimensional movement of airstreams within extra-tropical cyclones thus developing the conveyor belt model of extra-tropical cyclones. This model incorporates warm and cold conveyor belts which are now accepted and well studied aspects of extra-tropical cyclones (e.g. Thorncroft et al., 1993; Wernli and Davies, 1997; Eckhardt et al., 2004; Binder et al., 2016). Recently, the structure of intense extra-tropical cyclones in reanalysis data sets has been examined in a quantitative manner by creating cyclone composites (e.g. Bengtsson et al., 2009; Catto et al., 2010; Dacre et al., 2012). Cyclone composites have also been created using satellite observations of cloud fraction and precipitation (Field and Wood, 2007; Naud et al., 2010; Govekar et al., 2014; Naud et al., 2018) which enables cyclone structure in both reanalysis and model simulations to be systematically evaluated. Thus, considerable knowledge now exists of the spatial structure, dynamics and variability of the major precipitation producing air streams within extra-tropical cyclones.

A key question is then how the intensity, number, structure, and weather, for example precipitation, associated with extra-tropical cyclones, will change in the future as the climate warms. To answer this question, projections from climate models can be analysed. However, how the circulation responds to warming, which includes the characteristics of extra-tropical cyclones, is notably less clear and more uncertain than the bulk, global mean thermodynamic response (Shepherd, 2014). Champion et al. (2011) investigated the impact of warming on extra-tropical cyclone properties with one global climate model by comparing historical (1980-2000) simulations to future (2080-2100) simulations forced by the IPCC A1B scenario. They found small, yet statistically significant, changes to the 850-hPa maximum vorticity with the number of extreme cyclones increasing slightly in the future and the number of average intensity cyclones decreasing. Zappa et al. (2013b) analysed output from 19 models that participated in phase 5 of the Coupled Model Intercomparison Project (CMIP5) and compared 30-year periods of the historical (1976–2005) present-day simulations and the future climate simulations (2070–2099) forced by the representative concentration pathway 4.5 (RCP4.5) and 8.5 (RCP8.5) scenarios. In the RCP4.5 scenario, Zappa et al. (2013b) found a 3.6% reduction in the total number of extra-tropical cyclones in winter, a reduction in the number of extra-tropical cyclones associated with strong 850-hPa wind speeds, and an increase in cyclone related precipitation. In addition, they also note that considerable variability

in the response was found between different CMIP5 models. In a similar study, Chang et al. (2012) show that CMIP5 models predict a significant increase in the frequency of extreme extra-tropical cyclones during the winter in the Southern Hemisphere but a significant decrease in the most intense extra-tropical cyclones in winter in the Northern Hemisphere. A similar result was obtained by Michaelis et al. (2017) who used a mesoscale model to perform pseudo–global warming simulations over the North Atlantic where the initial and boundary conditions temperatures were warmed to a degree consistent with predictions from climate models forced with RCP8.5. They find a reduction in the number of strong storms with warming and an increase in cyclone precipitation.

Models participating in CMIP5 have systematic biases in the location of the climatological storm tracks in historical simulations, particularly in the North Atlantic where the storm track tends to be either too zonal or displaced southward (Zappa et al., 2013a). In addition, the response of the storm track to warming has been found to be correlated to the characteristics of the storm track in historical simulations. Chang et al. (2012) show that in the Northern Hemisphere individual models with stronger storm tracks in historical simulations project weaker changes with warming compared to individual models with weaker historical storm tracks. Moreover, the same study shows that in the Southern Hemisphere individual models with large equatorward biases in storm track latitude predict larger poleward shifts with warming.

In order to increase confidence in climate model projections of the number and intensity of extra-tropical cyclones there is a clear need to better understand the physical mechanisms causing changes to these weather systems. This is difficult to do based on climate model output alone as fully coupled climate models are very complex, include numerous feedbacks and non-linear interactions, and due to computational and data storage limitations offer somewhat limited model output fields with limited temporal frequency. Therefore in this study we undertake an idealised "climate change" experiment using a state-of-the-art model but configured as an aquaplanet.

Idealised studies have been used extensively in the past to understand the dynamics of extra-tropical cyclones. For example, baroclinic wave simulations have been performed to understand the dynamics of extra-tropical cyclones and fronts in the current climate (e.g. Simmons and Hoskins, 1978; Thorncroft et al., 1993; Schemm et al., 2013; Sinclair and Keyser, 2015). More recently baroclininc life cycle experiments have also been used to assess in a highly controlled simulation environment how the dynamics and structure of extra-tropical cyclones may respond to climate change. Given that diabatic processes, and in particular latent heating due to condensation of water vapour, play a large role in the evolution of extra-tropical cyclones (e.g. Stoelinga, 1996) many idealised studies have focused on how the intensity and structure of extra-tropical cyclones change as temperature and moisture content are varied (e.g Boutle et al., 2011; Booth et al., 2013, 2015; Kirshbaum et al., 2018). These studies show that when moisture is increased from low levels to values typical of today's climate, extra-tropical cyclones become more intense. This is a relatively robust result across many studies and can be understood to be a consequence of an induced low-level cyclonic vorticity anomaly beneath a localised maximum in diabatic heating (Hoskins et al., 1985). However, when temperatures and moisture content are increased to values higher than in the current climate, baroclinic life cycle experiments show divergent results. For example, Rantanen et al. (2019) found that uniform warming acts to decrease both the eddy kinetic energy and the minimum surface pressure of the cyclone whereas Kirshbaum et al. (2018) showed that for large temperature increases with constant relative humidity the eddy kinetic energy decreases whereas the minimum surface

pressure increases. Furthermore, Tierney et al. (2018) documented non-monotonic behavior of the cyclone intensity in terms of both maximum eddy kinetic energy and minimum mean surface pressure with increasing temperature.

A disadvantage of baroclinic life cycle experiments is that often only one cyclone and its response to environmental changes are considered whereas in reality there is considerable variability in the structure, intensity, size, and life time of extra-tropical cyclones. Recent baroclinic life cycle studies have suggested that the response of cyclones to warming in these type of simu-lations may depend on how the simulation is configured (Kirshbaum et al., 2018). An alternative, yet still idealised approach, is to perform multi-year aquaplanet simulations in which thousands of extra-tropical cyclones develop and can be analysed. A benefit of this approach compared to baroclinic life cycle experiments is that experimental set up and initial conditions have a much weaker influence of the evolution of the model state and thus on the structure and size of the simulated extra-tropical cy-clones. Pfahl et al. (2015) used a simplified general circulation model in an aquaplanet configuration with a slab ocean to assess how the intensity, size, deepening rates, life time, and spatial structure of extra-tropical cyclones respond when the longwave optical thickness is varied in such a way that the global mean near surface air temperature varies from 270 K to 316 K. Their main result was that changes in cyclone characteristics are relatively small except for the intensity of the strongest cyclones which increased in strength considerably with warming. However, this study was based on an idealized General Circulation Model which contained simplified physics parameterizations, for example, the large-scale microphysical parameterization only considers the vapor–liquid phase transition.

The first aim of this study is to determine how the number, intensity, and structure of extra-tropical cyclones change in response to horizontally uniform warming. The second aim is to identify the physical mechanisms which lead to changes in vertical motion and precipitation patterns associated with extra-tropical cyclones. These aims are addressed in an idealised modelling context as it is anticipated that mechanisms will be easier to identify than in complex, fully coupled climate model simulations. In particular, a full complexity atmospheric model is used to perform two aquaplanet simulations: a control sim-ulation and an experiment where the sea surface temperatures are uniformly warmed. Extra-tropical cyclones are then tracked and cyclone centred composites are created. The omega equation is used to determine the forcing mechanisms for vertical motion at different locations relative to the cyclone centre and at different points in the cyclone life cycle for extra-tropical cyclones in both the control and warm experiment.

The remainder of this paper is set out as follows. In section 2, the full-complexity numerical model, OpenIFS, which is used in this study is described along with the numerical experiments that are performed. In section 3, the cyclone tracking scheme and the omega equation diagnostic tool which are applied to the model output to assist with analysis are described. The results are presented in sections 4 to 7. The large-scale zonal mean state and its response to warming are given briefly in section 4 and the results concerning changes to bulk cyclone statistics are discussed in section 5. The results concerning changes to cyclone structure as ascertained from the cyclone composites are presented in section 6 and the impact of warming on the asymmetry of vertical motion in extra-tropical cyclones is considered in section 7. The conclusions are presented and discussed in section 8.

## 2 OpenIFS and numerical simulations

### 2.1 Numerical Model: OpenIFS

The numerical simulations are performed with OpenIFS which is a portable version of the Integrated Forecast System (IFS) developed and used for operational forecasting at the European Centre for Medium Range Forecasting (ECMWF). Since 2013, OpenIFS has been available under license for use by academic and research institutions. The dynamical core and physical parameterizations in OpenIFS are identical to those in the full IFS as are the land surface model and wave model. However, unlike the full IFS, OpenIFS does not have any data assimilation capacity. The version of OpenIFS used here (Cy40r1) was operational at ECMWF between November 2013 and May 2015. The full documentation of Cy40r1 is available online (ECMWF, 2015).

### 2.2 Experiments

Numerical simulations are performed with OpenIFS configured as an aquaplanet. The surface of the Earth is therefore all ocean and the sea surface temperatures (SSTs) are specified at the start of the simulation and held constant throughout the simulation. There is no ocean model included. However, the dynamics and physical parameterizations are exactly the same as in the full IFS and the wave model is also active in the aquaplanet simulations.

The control simulation (CNTL) is set up similarly to the experiments proposed by Neale and Hoskins (2000) and their QObs sea surface temperature distribution is used. This SSTs distribution is specified by a simple geometric function and is intended to resemble observed SSTs more so than the other distributions specified by Neale and Hoskins (2000). The resulting SST pattern is zonally uniform and symmetric about the equator. The maximum SST is 27°C in the tropics and poleward of 60°N in both hemispheres the SSTs are set to 0°C. There is no sea ice in the simulation. The atmospheric state is initialised from a randomly selected real analysis produced at ECMWF. First the real analysis is modified by changing the land-sea mask and setting the surface geopotential to zero everywhere. The atmospheric fields are then interpolated to the new flat surface in regions where there is topography on Earth. The perturbed experiment (hereinafter referred to as SST4) is identical to CNTL except that the SSTs are uniformly warmed by 4 K everywhere. Both experiments have a diurnal cycle in incoming solar radiation but no annual cycle; throughout the simulations the incoming solar radiation is fixed at the equinoctial value and is thus symmetric about the equator.

Both aquaplanet simulations are run at T159 resolution (approximate grid spacing of 1.125 degrees / 125 km) and with 60 model levels. The model top is located at 0.1 hPa. Both simulations are run for a total of 11 years and the first year of each simulation is discarded to ensure that the model has reached a balanced state. Analysis of global precipitation from the first year of simulation (not shown) reveals that a steady state is achieved after 3-4 months. Model output, including temperature tendencies from all physical parameterization schemes, is saved every 3 hours.

## 3 Analysis Methods

### 3.1 Cyclone Tracking and Compositing

In both numerical experiments, extra-tropical cyclones are tracked using an objective cyclone identification and tracking algorithm, TRACK (Hodges, 1994, 1995). Extra-tropical cyclones are identified as localised maxima in the 850-hPa relative vorticity truncated to T42 spectral resolution based on 6 hourly output from OpenIFS. All cyclones in the Northern Hemisphere are initially tracked, however to ensure that no tropical cyclones are included in the analysis, tracks which do not have at least one point north of 20°N are excluded. Furthermore, to ensure that only synoptic-scale, mobile systems are considered, it is required that a cyclone track must last for at least 2 days and travel at least 1000 km. Finally cyclones which have a maximum vorticity of less than $1 \times 10^{-5} \text{s}^{-1}$ are also excluded from the analysis. The output from TRACK consists of the longitude, latitude and relative vorticity value of each point (every 6 hours) along each individual extra-tropical cyclone track from which statistics such as genesis and lysis regions may be determined.

The cyclone tracks are then used as the basis to create composites of extra-tropical cyclones following the same method as Catto et al. (2010) and Dacre et al. (2012). Rather than creating a composite of all identified extra-tropical cyclones only the 200 strongest cyclones in terms of their maximum 850-hPa relative vorticity are selected from the CNTL and SST4 experiments and composites of a range of meteorological variables are created for these extreme cyclones at different offset times relative to the time of maximum intensity ($t = 0$ h). Each composite is created by first determining the values of the relevant meteorological variable, at each offset time, and for each individual cyclone to be included in the composite, on a spherical grid centered on the cyclone centre. The meteorological values are thus interpolated from the native model longitude-latitude grid to this spherical grid which has a radius of 12 degrees and is decomposed into 40 grid points in the radial direction and 360 grid points in the angular direction. To reduce smoothing errors, the cyclones are rotated so that all travel due east. To obtain the cyclone composite, the meteorological values on the radial grid are averaged at each offset time. Thus, the composite extra-tropical cyclone is the simple arithmetic mean of the 200 individual, rotated cyclones.

In addition to composites of the 200 strongest extra-tropical cyclones, composites of the 200 "most average" cyclones were also created for both the CNTL and SST4 experiments. These cyclones were identified as the 100 cyclones with maximum vorticity values lower but closest to the median relative vorticity and the 100 cyclones with maximum vorticity values higher than but closest to the median relative vorticity. The results from these median composites are shown in the supplementary material as although uniformly warming the SSTs lead to increases in the total column water vapour and precipitation it had little coherent impact on the spatial structure of the median extra-tropical cyclone.

### 3.2 Omega Equation

The omega equation is a diagnostic equation from which the vertical motion ($\omega$) resulting from different physical processes can be calculated. Different forms of the omega equation with differing degrees of complexity exist and range from the simplest "standard" Quasi-Geostrophic (QG) form with friction and diabatic heating neglected (Holton and Hakim, 2012) to the complex generalized omega equation (Räisänen, 1995; Rantanen et al., 2017). Here we solve the following version of the QG omega

equation in pressure ($p$) coordinates:

$$\sigma_0(p)\nabla^2\omega + f^2\frac{\partial^2\omega}{\partial p^2} = f\frac{\partial}{\partial p}\left(\boldsymbol{v}\cdot\nabla\left(\zeta+f\right)\right) + \frac{R}{p}\nabla^2\left(\boldsymbol{v}\cdot\nabla T\right) + \frac{R}{c_p p}\nabla^2 Q. \tag{1}$$

The static stability parameter, $\sigma_0$, is only a function of pressure and time and is given by

$$\sigma_0(p) = \frac{-RT_0}{p}\frac{1}{\theta_0}\frac{\partial\theta_0}{\partial p} \tag{2}$$

where $\theta_0$ ($T_0$) is the horizontally averaged potential temperature (temperature) profile over the global domain calculated at every time step. The left-hand side operator of Eq. (1) is identical to the standard QG omega equation. The terms on the right-hand side of Eq. (1) represent forcing for vertical motion due to differential vorticity advection, thermal advection and diabatic heating respectively. The right-hand side differs from the standard QG omega equation in that diabatic heating ($Q$) is retained, the advection terms are calculated using the full horizontal winds ($\boldsymbol{v}$) rather than the geostrophic winds, and the full relative vorticity ($\zeta$) is advected rather than the geostrophic vorticity. Friction is neglected as on an aquaplanet this is expected to be small.

Overall good agreement is found between the model calculated vertical motion and the vertical motion diagnosed by Eq (1). Correlation coefficients between the model calculated vertical motion and the diagnosed vertical motion were calculated at each grid box and pressure level and averaged over latitude bands (not shown). In the latitude band 30–60°N the correlation coefficients were 0.84 at 700 hPa and exceeded 0.9 at 500 hPa.

## 4 Climatology and large-scale response to warming

In this section the zonal mean climatology of CNTL is described along with the response to the uniform warming. Figure 1 shows that the control simulation produces a realistic distribution of temperature and of zonal winds. The dynamic tropopause varies from 300 hPa in the polar regions to about 100 hPa in the tropics, similar to what is observed on Earth. The zonal mean jet streams have maximum wind speeds of 45 m s$^{-1}$ and are located on the tropopause at 35°N/S. As expected from the aquaplanet model set-up the two hemispheres are almost symmetrically identical.

The response to the uniform 4 K warming is shown by the shading in Fig. 1. The temperature increases everywhere in the troposphere with the largest warming in the tropical upper troposphere, where temperature increases by up to 7 K. Cooling takes place in the polar stratosphere which acts to increase the upper-level meridional temperature gradient. The tropopause height increases at most latitudes with warming. The spatial pattern of these changes in zonal mean temperature are similar to those found in more complex climate models (e.g. Figure 12.12, Collins et al., 2013). However, the warming in the low to mid troposphere is relatively uniform with latitude. The lack of enhanced warming in the Northern Hemisphere polar regions (polar amplification) and hence no decrease in low-level baroclinicity is the most notable difference in the atmosphere's response to warming in these aquaplanet experiments compared to in complex climate model simulations.

At low-levels, the increase in temperature in the SST4 experiment relative to CNTL is typically of order 4 K which is of similar magnitude to the enforced increase in SSTs. This temperature increase can be put into context by comparison

with predictions from CMIP5 models. Under the RCP8.5 scenario, CMIP5 models predict that global mean near surface temperatures will increase by 2.6 to 4.8 K by the end of the 21st century relative to the 1986–2005 mean. Hence, the aquaplanet simulations performed here have a degree of warming that could be expected to occur by the end of 21st century under large greenhouse gas emissions.

The response of the zonal mean zonal wind shows that the sub-tropical jet intensifies and moves vertically upwards. The eddy-driven jet, evident at low-levels, displays a dipole structure indicative of a poleward shift. This is confirmed when the latitude of the maximum 700 hPa zonal mean zonal wind speed is considered: this moves polewards by 3.3° in the SST4 experiment compared to in CNTL. These responses of the zonal mean jet streams to uniform warming are similar to those found in more complex climate models (e.g. Collins et al., 2013), particularly in the Southern Hemisphere, demonstrating that the OpenIFS aquaplanet can realistically simulate an Earth-like atmosphere.

The zonal mean precipitation in both CNTL and SST4 experiment are shown in Fig. 2a. Again strong similarities exist with real Earth observations and CMIP5 model projections (e.g. Lau et al., 2013). The largest rainfall is observed in the tropics and a secondary peak occurs in the mid-latitudes which is associated with the mid-latitude storm track. The effect of warming the SSTs is to increase the mean precipitation at almost all latitudes. The largest absolute increase occurs in the tropics. In the Northern Hemisphere mid-latitudes the maximum precipitation rate increases from 3.9 to 4.2 mm day$^{-1}$ and the location of the maximum moves polewards by 2.2 degrees. This is in agreement with the poleward shift in the eddy driven jet and strongly suggests that, on average, extra-tropical cyclones move poleward with warming. This will be confirmed in Sect. 5.

Figure 2b shows the zonal mean mean sea level pressure (MSLP). The highest zonal mean MSLP in the CNTL experiment occurs in the subtropics and moves poleward with warming. The lowest values of MSLP occur on the poleward side of the jet stream and again move poleward with warming. A notable difference between these MSLP distributions and the MSLP distribution on Earth is the absolute magnitude of the values. The mean MSLP on Earth is 1013 hPa whereas in both the CNTL and SST4 experiments, the global mean MSLP is 985.4 hPa. This difference is solely due to the initialisation method (see Sect. 2.2) and the average surface pressure of 985.4 hPa results, as it is the average pressure at the actual surface height in the randomly selected analysis used for the initialisation. Figure 2c shows the zonal mean 950-hPa temperature which indicates that the low-level temperature increase is almost constant with latitude and implies that the low-level baroclinicity does not change.

The impact of uniformly warming the SSTs on the baroclinicity can be quantified via the maximum (dry) Eady growth rate, $\sigma$, which is given by

$$\sigma = 0.31 \frac{|f|}{N} \left| \frac{\partial u}{\partial z} \right|, \tag{3}$$

where $f$ is the Coriolis parameter, $N$ is the Brunt-Väisälä frequency and $u$ is the zonal wind component. In the CNTL simulation (Fig. 3a) the Eady growth rate has maximum values of 0.75 day$^{-1}$ in the mid-latitude mid-to-upper troposphere. A secondary maximum is evident in the stratosphere, however, this most likely has little significance for the growth of extra-tropical cyclones. The response of the Eady growth rate to warming includes an increase just above the dynamical tropopause and a decrease co-located with the secondary maximum in the stratosphere. With the mid-troposphere, the Eady growth rate

decreases slightly with warming, for example, at 700 hPa the maximum value decreases from 0.54 day$^{-1}$ in CNTL to 0.50 day$^{-1}$ in SST4. Close to the surface, at 900 hPa, the maximum value of the Eady Growth rate also experiences a small decrease with warming, from 0.92 day$^{-1}$ to 0.89 day$^{-1}$. The most notable impact of warming on the Eady growth rate at 900 hPa is a poleward shift of 5.4 degrees in the position of the maximum. Equatorward of 45°N the 900-hPa Eady growth rate decreases with warming whereas poleward of 45°N it increases.

Figures 3b and c show the vertical shear of the zonal wind and the Brunt-Väisälä frequency respectively. There is little change in the vertical wind shear with warming in the mid-tropophere, which via thermal wind balance is consistent with the lack of any large changes to the horizontal temperature gradient in the troposphere (Fig. 1a). Near the surface, there is a dipole pattern showing that the maximum in wind shear moves polewards. This is consistent with the poleward shift of the eddy-driven jet and also explains the poleward shift in the 900-hPa Eady growth rates. The Brunt-Väisälä frequency increases in the troposphere which indicates that the decrease in the Eady growth rates at 700 hPa, and, at lower latitudes, higher up in the troposphere, is due primarily to changes in the static stability. Near the tropopause the decrease in the stability associated with an increase in the tropopause height increases the Eady growth rate. In contrast, the decrease in the secondary maximum in the stratosphere is due to changes in the vertical wind shear.

## 5 Cyclone Statistics

In this section bulk cyclone statistics are presented from both the CNTL and SST4 simulations. All cyclone tracks that meet the criteria described in Sect. 3.1 are included in this analysis and their mean and median characteristics are summarised in Table 1. In the control simulation there are 3581 extra-tropical cyclones which have a median life time of 108 hours (4.5 days) and a median maximum vorticity of $5.94 \times 10^{-5}$ s$^{-1}$ (Table 1). The uniform warming acts to decrease the total number of cyclone tracks by 3.3% but does not alter the median duration (life time) of extra-tropical cyclones (Table 1). The inter-annual variability in the number of cyclone tracks, quantified by calculating the number of cyclone tracks each year and then obtaining the standard deviation of these 10 values, is small (13.5 in CNTL and 10.1 in SST4) relative to the absolute decrease in the number of cyclone tracks (119). This, and a two-sided Student's t-test, show that the decrease in the number of tracks is statistically significant.

Figure 4a shows histograms of maximum 850-hPa vorticity (also referred to hereinafter as maximum intensity). There are more stronger cyclones, for example with intensities exceeding $10 \times 10^{-5}$ s$^{-1}$, in the SST4 experiment than in the CNTL experiment. However, the mean intensity does not change considerably and there is a 3.2% decrease (equivalent to $0.19 \times 10^{-5}$ s$^{-1}$) in the median maximum vorticity. This change (i.e. the signal) is very small compared to the variation between individual cyclones quantified by the standard deviation of the maximum relative vorticity of all storms ($2.55 \times 10^{-5}$ s$^{-1}$ in CNTL). Furthermore, the mean maximum vorticity for all cyclones occurring in each individual year can be obtained and the standard deviation of these ten values calculated to obtain the inter-annual standard deviation of the maximum relative vorticity. For CNTL this is $0.14 \times 10^{-5}$ s$^{-1}$ and $0.10 \times 10^{-5}$ s$^{-1}$ in SST4 which are both larger than the absolute change in the mean maximum relative vorticity (-0.04 $\times 10^{-5}$ s$^{-1}$, Table 1). A two-sided student t-test further confirms that the mean

intensity does not differ in a statistically significant way between CNTL and SST4 and a Wilcoxon rank sum test shows that
the median maximum intensities are not statistically significantly different between the CNTL and SST4 experiment. However,
as evident from Fig. 4a, and confirmed by a one tailed F-test applied to the maximum vorticity distributions, the maximum
vorticity distribution in the SST4 experiment has a larger variance than in the CNTL experiment. Thus, it can be concluded that
the average population of all cyclones does not change with warming but that there are more stronger, and more weak cyclones
in the SST4 experiment than in CNTL.

Table 1 also includes the median deepening rates of all extra-tropical cyclones. The deepening rate is the temporal rate of
change of the 850-hPa relative vorticity so that positive values indicate a strengthening, or deepening, of the extra-tropical
cyclone. In CNTL, the relative vorticity increases by $1.31 \times 10^{-5}$ $s^{-1}$ every 24 h when evaluated from genesis time to time of
maximum intensity. In SST4, the corresponding value is $1.28 \times 10^{-5}$ $s^{-1}$ per 24 h. The change is very small in comparison
to the standard deviation of the deepening rates (Table 1). Distributions of the deepening rates of all identified extra-tropical
cyclones calculated between the time of genesis and time of maximum intensity are shown in Fig. 4b. A ranksum test performed
on the deepening rates between the time of genesis and time of maximum intensity confirms that the median values are not
statistically different. The same test applied to the deepening rates calculated over the 24 hours before the time of maximum
intensity also shows that the control and SST4 experiments do not differ significantly. However, similar to what is found
with the maximum vorticity distributions, the variance of the deepening rates is statistically significantly larger in the SST4
experiment compared to in CNTL. The lack of any notable change in the median deepening rate of all extra-tropical cyclones
differs somewhat from the zonal mean calculations of the Eady growth rate (Fig. 3a) which indicate a 5-10% decrease. This
difference likely arises because the Eady growth rate is a measure of dry baroclinicity whereas moist processes are acting in
these simulations.

Distributions of the genesis and lysis latitudes for all extra-tropical cyclones are shown in Fig. 4c and d. As hypothesised in
Sect. 4 both genesis and lysis regions move poleward with warming. The median genesis region moves 2 degrees polewards
from 44.2°N to 46.2°N and the median lysis region moves poleward by 1.9 degrees from 51.4°N to 53.3°N (Table 1). The
inter-annual standard deviation of the genesis latitude is 0.27° in CNTL and 0.69° in SST4 suggesting that the two degree
poleward shift in genesis latitude is significant. Likewise, the inter-annual standard deviation of the lysis latitude are 0.52° and
0.72° in CNTL and SST4 respectively and therefore also smaller than the response to warming. Two-sided student t-tests show
that the mean genesis latitude differs between the CNTL and SST4 experiments at the 95% confidence level and that both
the median genesis and lysis latitudes differ significantly at the 0.05 significance level. The standard deviation of the both the
genesis latitude (12.8° in CNTL, Table 1) and lysis latitude (13.7° in CNTL, Table 1) of all cyclones is larger than the mean
change in genesis and lysis latitudes indicating that the change is small compared to the variation between individual cyclones.

Table 1 also includes statistics for the strongest 200 extra-tropical cyclones in each experiment as the structure of these
intense extra-tropical cyclones will be investigated in detail in Sect. 6. Firstly, the median genesis latitudes of the strongest
extra-tropical cyclones are 6 to 8 degrees farther equatorward than for all extra-tropical cyclones in both the CNTL and SST4
experiments which means that the strongest storms form in climatologically warmer and more moist environments than average
intensity storms. The more equatorward genesis region, combined with similar (CNTL) or more poleward (SST4) lysis regions

means that the strongest extra-tropical cyclones have much larger latitudinal displacements than extra-tropical cyclones do on average. Secondly, the median genesis latitude of the 200 strongest extra-tropical cyclones only moves 0.4 degrees poleward with warming which is notably less than the 2.0 degree poleward shift found when all extra-tropical cyclones are considered. Thirdly, deepening rates increase much more with warming for the strongest 200 extra-tropical cyclones than for all extra-tropical cyclones. Finally, the mean, median and maximum intensity of the 200 strongest extra-tropical cyclones in the SST4 experiment are larger than in the CNTL experiment.

## 6  Cyclone Structure

### 6.1  Evolution of the composite cyclone in CNTL

The cyclone composite of the strongest 200 extra-tropical cyclones in the CNTL experiment is now discussed. The temporal evolution of the composite mean cyclone in the CNTL simulation, in terms of mean sea level pressure and total column water vapour (TCWV) is shown in Fig. 5. Forty-eight hours before the time of maximum intensity ($t$ =-48 h, Fig. 5a) the composite cyclone has a closed low pressure centre with a minimum MSLP of 978 hPa. The location of the cold and warm fronts are evident as enhanced gradients in the TCWV and the warm sector, located between the cold front and warm front, is well defined and has values of TCWV exceeding 25 g kg$^{-1}$. At 24 hours before the time of maximum intensity ($t$ =-24 h, Fig. 5b) the low pressure centre has become deeper (minimum MSLP of 960 hPa), the warm sector has become narrow and the gradients in TCWV across both the warm and cold fronts have become larger. The dry air moving cyclonically behind the cold front, now extends farther south relative to the cyclone centre than it did 24 hours earlier. By the time of maximum relative vorticity ($t$ =0 h, Fig, 5c), the MSLP shows a mature, very deep cyclone which has a minimum pressure of 944 hPa. The TCWV over the whole cyclone composite area is now considerably lower than at earlier stages most likely because as the cyclones included in the composite intensify they move poleward to climatologically drier areas. The TCWV pattern also shows that the composite cyclone starts to occlude by this point ($t$ =0 h) as the warm sector does not connect directly to the centre of cyclone - instead it is displaced downstream. Finally 24 hours after the time of maximum intensity ($t$ =+24 h, Fig. 5d), the cyclone resembles a barotropic low and has weak frontal gradients associated with it. The evolution of the composite cyclone in the CNTL experiment is, however, qualitatively very similar to real extra-tropical cyclones observed on Earth.

### 6.2  Low-Level Potential and Relative Vorticity

The response of the cyclone structure to warming is now considered primarily using changes to the mean values (i.e SST4 - CNTL). First the temporal evolution of the low-level potential vorticity (PV) and the changes to this variable with warming are considered (Fig. 6). Before the composite cyclone reaches it maximum intensity (Figures 6a,b), the maximum in the 900–700 hPa layer-averaged PV in the control simulation is poleward and downstream of the cyclone centre. By the time of maximum intensity (Fig. 6c), the maximum PV is co-located with the cyclone center and there is a secondary maximum which extends downstream of the cyclone centre and is co-located with the warm front.

At $t$ =-48 h and $t$ =-24 h, the largest absolute increases in the 900–700 hPa PV occurs poleward of the warm front location (Fig. 6a,b). This low-level PV anomaly is primarily caused by a diabatic heating maximum above this layer and therefore the poleward movement of the maximum indicates that the maximum in diabatic heating has also moved polewards with warming. The increase in PV is co-located with an increase in relative vorticity (orange contours in Fig. 6) which is consistent with an intensified cyclonic circulation beneath a region of localised heating. It can therefore be concluded that the relative vorticity associated the warm front increases with warming. At $t$ =0 h and $t$ =+24 h (Fig. 6c,d), two distinct regions of increased low-level PV are evident. The first is poleward of the warm front, as found at the earlier stages of development, and the second is almost co-located with the cyclone centre yet displaced slightly downstream. Both localised increases in low-level PV are also associated with increases in relative vorticity. In relative terms (not shown) the low-level potential vorticity poleward of the warm front increases by 25-30%, whereas near the cyclone centre the low-level PV only increases by 15–20%.

The response to warming also shows that almost everywhere within a 12 degree radius of the cyclone centre, at all offset times, there is an increase in low-level PV. The absolute values of increase are smaller, mostly less than 0.1 PVU but in relative terms the increase is similar in magnitude to that found near the warm front and cyclone centre. Away from the cyclone centre, where there is no significant relative vorticity, this increase in low-level PV is primarily caused by an increase in stratification. However, given that the cyclones are more poleward in the SST4 experiment the increase in planetary vorticity also plays a small role.

### 6.3   Low-level Wind Speed

Figure 6 highlighted that the relative vorticity increases with warming at all offset times. Associated with this increase in relative vorticity is an increase in low-level horizontal wind speeds. In the composite from the CNTL experiement, at $t$ =-48 h and $t$ =-24 h (Fig. 7a, b), the strongest 900-hPa wind speeds exceed 20 ms$^{-1}$ and 25 ms$^{-1}$ respectively and occur in the warm sector. At the time of maximum intensity (Fig. 7c), the strongest 900-hPa winds in CNTL are located equatorward of the cyclone centre, behind the cold front in a very dry area and exceed 30 ms $^{-1}$. By $t$ =24 h (Fig. 7d) the wind speeds have started to weaken. At $t$ =-48 h, a dipole structure in the change in wind speed due to warming is evident indicating that the maximum wind speeds move poleward and downstream relative to those in CNTL (Fig. 7a). However, the positive values are greater in magnitude than the negative values thus indicating an overall increase in wind speed. At $t$ =-24, 0 and 24 h (Fig. 7b–d) the 900-hPa winds speeds of the composite cyclone increase with warming by $\sim$ 1.5 ms$^{-1}$ in a large area surrounding the cyclone and by up to 3.5 ms$^{-1}$ in the warm front area. Consequently, the size of the area affected by wind speeds over a fixed threshold value increases indicating greater wind risk in warmer climates. As the increase at all offset times is not co-located with maximum wind speed in CNTL, this suggests that the spatial structure of the composite extra-tropical cyclone changes with warming.

### 6.4   Total Column Water Vapour

The response of the TCWV is now considered (Fig. 8). The uniform warming leads to an increase in TCWV everywhere in the cyclone composite at all off-set times. The largest absolute increases occur at $t$ =-48 h and $t$ = -24 h (Fig. 8a,b). At both

of these offset times, the largest absolute increase occurs in the warm sector where the mean values are largest in the control simulation. In terms of percentage increase (not shown), at $t =$-24 h, the TCWV increases the least, approximately 25%, in the cold sector upstream of the cyclone centre and the most ahead of the warm front where the increase exceeds 50%. At the time of maximum intensity (Fig. 8c), absolute increases of up to 6 g kg$^{-1}$ are still evident in the warm sector and in a localised region north-east of the cyclone centre whereas at $t =$+24 h (Fig. 8d) increases of this magnitude are constrained to the most southern part of the cyclone composite. The composites also show the meridional moisture gradient across the composite cyclone increases notably with warming since the absolute increase is much larger in the most equatorward regions (e.g. 12 g kg$^{-1}$ at $t =$-48 h) than in the most poleward regions (e.g an increase of 2 g kg$^{-1}$).

## 6.5  Precipitation

The response of the total, convective, and large-scale precipitation to warming is now considered. Composites of total, large-scale and convective precipitation are shown in Fig. 9 valid 48, 24 and 0 hours before the time of maximum intensity. Precipitation is calculated as the 6-hour accumulated value centred on the valid time in units of mm (6 h)$^{-1}$. In the CNTL simulation the maximum total precipitation is downstream and poleward of the cyclone centre at all offset times. At $t =$-48 h, the total precipitation has maximum values of 6 mm (6 h)$^{-1}$ and is mainly located in the warm sector of the cyclone and near the warm front (Fig. 9a). At $t =$-24 h, the total precipitation in the CNTL simulation is slightly larger, covers a greater area and has a more distinct comma shape than 24 hours earlier (Fig. 9d). Also at this time, large values of total precipitation are evident along the cold front to the south of the cyclone centre. By the time of maximum intensity the total precipitation in the CNTL experiment has started to decrease, with maximum values of 4 mm (6 h)$^{-1}$, and the location of the precipitation has rotated cyclonically around the cyclone centre (Fig. 9g).

The response to warming of the total precipitation is a large absolute and relative increase at all offset times. The maximum absolute increases are of order 2.5, 3.5 and 2.0 mm (6 h)$^{-1}$ at $t =$-48 h, $t =$-24 h, and the time of maximum vorticity ($t =$0 hr) respectively. These values correspond to relative increases of up to almost 50%. The maximum increase in the total precipitation is not co-located with the maximum in the CNTL simulation indicating that the spatial structure of the composite cyclone has changed with warming.The largest increases in total precipitation occurs in the warm front region, poleward and downstream of the maximum in the CNTL simulation at all offset times. This spatial change is largely similar to that found when the 900–700 hPa potential vorticity response to warming was considered (Fig. 6). This is consistent in the sense that more precipitation, and particularly more condensation, results in more latent heating and thus a stronger positive PV anomaly beneath the localised heating. However at $t =$0 h, it is interesting to note that while there is only one localised area where precipitation increases in SST4 compared to in CNTL (Fig. 9g), which is ahead of the warm front, there are two regions where the low-level PV increases (Fig. 6c). One of these regions is co-located with the increase in precipitation but the second region is closer to the cyclone centre. While this may be due to the larger mean relative vorticity of the strongest 200 cyclones in SST4 compared to CNTL (Table 1), it is also possible that this second area of enhanced PV may be due to enhanced advection by the cold conveyor belt of PV produced diabatically in the warm front region, beneath the ascending warm conveyor belt. Schemm and Wernli (2014) noted such a mechanism in their study linking warm and cold conveyor belts.

The contribution of the large-scale stratiform precipitation calculated from the cloud scheme and the convective precipitation produced by the convection scheme to the total precipitation is now considered. In CNTL, the large-scale precipitation (Fig. 9b, e, h) contributes more to the total precipitation than the convective precipitation (Fig. 9c, f, i) particularly at $t =$-24 hrs and the time of maximum intensity. However, the convective precipitation is larger and of equal magnitude to the large-scale precipitation in the more equatorward parts of the warm sector of the CNTL composite cyclone where the temperature and moisture content are higher. The large-scale precipitation increases in SST4 compared to CNTL in the warm frontal region, poleward of the maximum in the CNTL simulation, at all offset times. This spatial shift is very similar to that observed for the total precipitation meaning that the resolved precipitation is leading to the poleward shift in the total precipitation with warming. However, the large-scale precipitation also has a smaller increase (1–1.5 mm per 6 hr) in a narrow band along the cold front, upstream of the maximum in the control simulation which is most evident at $t =$-24 hrs. In contrast, the convective precipitation, which increases by almost 50%, has the largest increases co-located with the maximum in the control simulation, meaning that the position of convective precipitation relative to the cyclone centre does not change with warming.

## 6.6  Vertical Velocity

The mean cyclone composite of vertical velocity at 700 hPa (given in pressure coordinates, Pa s$^{-1}$) obtained directly from the model simulations and the response to warming is shown in Fig. 10a, c and e. In the CNTL simulation at $t =$-48 h and $t =$-24 h, there is large coherent area of ascent downstream of the cyclone centre largely co-located with the warm sector indicative of the warm conveyor belt, an ascending air stream associated with extra-tropical cyclones. At $t =$0 h (Fig. 10e), the area of ascent is still maximised in the warm sector region but is further downstream relative to the cyclone centre than at earlier times. The ascent at $t =$0 h has also started to wrap cyclonically around the poleward and upstream side of the cyclone meaning that the cyclone has formed a bent-back warm front and likely has started to occlude. The absolute magnitude of the largest values of ascent occur at $t =$-24 h and exceed 0.6 Pa s$^{-1}$ (Fig. 10c), approximately 6 cm s$^{-1}$. A region of weak descent is evident behind the cold front in the drier air mass at all offset times.

Uniform warming changes the vertical motion in a complex manner. The largest increases in ascent are not co-located with the strongest ascent in the CNTL simulation and instead occur poleward and downstream of the maximum. This pattern is present at all offset times and suggests that the warm front and the warm conveyor belt are located farther poleward relative to the cyclone centre in the SST4 simulation. This is consistent with the response of the total and large-scale precipitation, and the low-level potential vorticity which also showed a poleward shift in the warm frontal region. A tri-pole structure is also evident in Fig. 10a, c and e which shows that the area of ascent either weakens in the centre and broadens with warming or that the ascent associated with the warm and cold fronts become more spatially separate with warming. The first of these two options will prove to be correct.

To further understand the spatial pattern of the response of the vertical velocity to warming, the contribution to the total vertical velocity from vorticity advection, thermal advection and diabatic processes as diagnosed by the omega equation (Eq. (1)) are examined. The sum of these three terms (Fig. 10b, d, f) at 700 hPa is first compared to the total, model calculated vertical motion (Fig. 10a, c, e). At $t =$-48 h, the diagnosed ascent in CNTL is slightly weaker than the model calculated (i.e direct from

OpenIFS) ascent particularly in the cold front region. The response of the diagnosed vertical motion to warming is however spatially similar to that of the model calculated vertical motion. At $t =$-24 h, the diagnosed ascent is slightly stronger than the model calculated ascent and covers a larger area especially in the zonal direction. In addition, the descent diagnosed from Eq. (1) covers a smaller area than descent in the model calculated vertical motion field. Similar differences between the model calculated and diagnosed vertical motion occur at $t =$0 h. There is, however, broad agreement between the model calculated vertical motion and the vertical motion diagnosed using Eq. (1) in CNTL at all offset times and the response to warming in the diagnosed vertical motion field is very similar to that in the model calculated field. Thus, the individual contributions to the diagnosed ascent will provide reliable estimates of how different physical processes influence the total vertical motion.

Figure 11 shows the contribution to vertical velocity from thermal advection, vorticity advection and diabatic heating at three offset times in the CNTL experiment. The maximum values of ascent attributed to the different forcing mechanisms in the composite mean cyclones are also shown in Table 2. In the CNTL composite cyclone thermal advection leads to ascent in the warm sector downstream of the cyclone centre and descent behind the cold front (Fig. 11a, d, g). Ascent and descent due to thermal advection reach a maximum 24 hours before the time of maximum intensity (Table 2) and as the composite cyclone evolves in time the region of ascent wraps cyclonically around the poleward side of the cyclone centre. Vorticity advection (Fig. 11b, e, h) leads to stronger ascent at all offset times compared to either thermal advection or diabatic heating in the CNTL composite cyclone (Fig. 11, Table 2). At all offset times the ascent attributed to vorticity advection covers a large area and is located downstream of the cyclone centre. As the cyclone evolves, the area of maximum ascent moves further away from the cyclone centre and a region of weak ascent wraps cyclonically around the cyclone centre. This cyclonic behaviour, indicative of lifecycle 2 (LC2, Thorncroft et al. (1993)) cyclone development, occurs as the cyclones are located on the poleward side of the jet at all offset times considered here. The contribution to vertical motion from the diabatic heating term is shown in Figs. 11c, f, and i. Ascent related to diabatic heating is constrained to a smaller area than ascent due to either vorticity or thermal advection and is located in the poleward parts of the warm conveyor belt. The maximum values of ascent related to diabatic heating at $t =$-24 h and the time of maximum intensity are also weaker than those due to either thermal advection or vorticity advection in both the CNTL and SST4 experiments (Table 2). This shows that at 700 hPa diabatic heating has a smaller impact on the cyclone's vertical motion than the dynamical terms.

The response of vertical motion due to the different forcing mechanisms to warming is also shown in Fig. 11 by the shading. At $t =$-48 h, ascent due to thermal advection weakens slightly with warming (orange shading co-located with the maximum ascent in CNTL in Fig. 11a and Table 2) and the descent associated with cold-air advection is also weaker in SST4 than in the CNTL. The ascent due to warm-air advection in the warm sector is slightly more poleward in SST4 compared to CNTL. At $t =$-24 h, the region of ascent due to thermal advection in SST4 has moved poleward and downstream relative to that in the CNTL. This is evident in Fig. 11d as the positive difference values (weaker ascent, orange shading) between the cyclone centre and the maximum ascent in the CNTL and as negative difference values (stronger ascent, purple shading) poleward of the maximum ascent in CNTL. This illustrates that at both $t =$-48 h and $t =$-24 h the warm front is more poleward and extends further downstream away from the cyclone centre in SST4 compared to in CNTL. A similar, but weaker pattern also remains at the time of maximum intensity (Fig. 11g).

The response of vertical motion due to vorticity advection at 700 hPa has a similar spatial pattern at all offset times considered but is most pronounced in magnitude at $t =$-24 hours and at the time of maximum intensity. The most notable feature is that ascent due to vorticity advection in SST4 covers a greater area compared to in CNTL and that the ascent expands polewards and downstream of the cyclone centre. The second notable feature is that at 700 hPa the maximum ascent due to vorticity advection decreases with warming (Fig. 11b, e, h and Table 2). This indicates that ascent due to positive vorticity advection downstream of the cyclone centre weakens in magnitude but becomes more spatially extensive.

To further understand the change in ascent due to vorticity advection, the 500 hPa geopotential height fields are considered. To compare the 500 hPa geopotential height in the SST4 and CNTL composite cyclones, first the composite cyclone mean (weighted by grid area) at each offset time was subtracted to generate maps of the cyclone relative 500 hPa geopotential height anomaly. This was necessary as in SST4 the 500 hPa heights are higher simply due to the warmer atmosphere which makes a comparison of the shape and extent of the upper level trough difficult. Figure 12 shows the differences in these anomalies. At $t =$-48 h (Fig. 12a) the negative anomaly to the south of the cyclone centre indicates that the 500-hPa trough is slightly deeper in SST4 compared to in CNTL. Furthermore, the dipole of negative and positive anomalies downstream of the cyclone centre at $t =$-48 h indicates that the 500-hPa trough is sharper in SST4 compared to in CNTL. At $t =$-24 h (Fig. 12b) an asymmetric dipole pattern is evident which has small positive values upstream and larger negative values downstream of the cyclone centre. This indicates that the 500-hPa trough is shifted downstream relative to the cyclone centre in SST4 compared to in CNTL but also that the trough is broader and extends more downstream in SST4 compared to CNTL. This pattern is also evident at $t =$0 hrs and $t =$+24 hrs. The broader upper level trough in SST4 is thus the likely reason why ascent due to vorticity advection expands over a greater area downstream in SST4 compared to CNTL.

Ascent attributed to diabatic heating has a larger relative increase with warming in the warm front region compared to both dynamical terms (Fig. 11). At all offset times, ascent due diabatic heating increases poleward of the maximum in the CNTL composite, which combined with the absence of any decrease in ascent results in an expansion of the area where diabatic heating is contributing notably to ascent. Furthermore, in contrast to both thermal advection and vorticity advection there are no coherent regions where descent due to diabatic heating has increased.

The spatial patterns of changes in ascent due to the different forcing mechanisms (Fig. 11) can be compared to the patterns of change in the total, model output vertical velocity (Fig. 10). It can therefore be concluded that the increase in ascent poleward and downstream of the cyclone centre occurs due to a combination of all three processes. However, thermal advection and diabatic heating are responsible for most of the increase in ascent close to the cyclone centre whereas vorticity advection is the main cause of the downstream expansion of the ascent field. The decrease in total ascent near the cyclone centre is found to be due to changes in spatial pattern of thermal advection and the position of the fronts. In the SST4 experiment, the warm front advances further ahead of the cyclone centre than in the CNTL which results in weaker ascent due to warm air advection close to the cyclone centre. However, the decrease in ascent in the warm sector is a direct result of weaker ascent due to vorticity advection in this location which arises as a consequence of the broader 500-hPa trough. In contrast, the weaker descent immediately to the south of the cyclone centre is mainly due to weaker cold-air advection in SST4 compared to CNTL

which again relates to changing positions of the fronts. Finally, it should be noted that vertical velocities are likely to be weaker for the same forcing in the warmer experiment due to an increase in tropospheric static stability (e.g. Fig. 3c).

## 7 Asymmetry of vertical motion

Ascent is stronger and occupies a smaller and narrower region than descent, both in the context of extra-tropical cyclones and globally. O'Gorman (2011) proposed two parameters, $\lambda$ and $\lambda_{Area}$, to quantify the asymmetry between upward and downward motions. The asymmetry parameter, $\lambda$ is given by

$$\lambda \equiv \frac{\overline{\omega'\omega^{\uparrow\prime}}}{\overline{\omega'^2}} \tag{4}$$

where in our study we define $\omega'$ to be the perturbation from the mean vertical velocity (Pa s$^{-1}$) calculated over the 12 degree cyclone composite area. $\omega^{\uparrow}$ denotes taking the upward part of $\omega$ (i.e. applying a Heaviside function). The overbar denotes an average over the cyclone area. $\lambda_{Area}$ is an alternative (and more approximate) definition of asymmetry which is based on geometric considerations. $\lambda_{Area}$ is given by

$$\lambda_{Area} = \frac{1 - a_u}{1 - \overline{\omega}/\overline{\omega_u}} \tag{5}$$

where $a_u$ is the fraction of the total area (in this case the 12 degree cyclone composite area) which is covered by ascent, $\overline{\omega_u}$ is the spatial average over the cyclone area of the ascent and $\overline{\omega}$ is the spatial average of all vertical motions.

$\lambda$ and $\lambda_{Area}$ are both calculated for each of the 200 individual extra-tropical cyclones in both the CNTL and SST4 experiments at different offset times. Only vertical motions at one pressure level, 700 hPa, are considered. Figure 13 shows the distribution of both asymmetry parameters at $t =$-24 h which reveals that there is considerable spread across the 200 strongest extra-tropical cyclones particularly in $\lambda$ with values ranging from 0.55 to 0.9 in both experiments. However, it is also evident that there are larger values of both $\lambda$ and $\lambda_{Area}$ in SST4 compared to in CNTL. The mean values for both parameters are shown in Table 3. At $t =$-24 h, $t =$0 h and $t =$+24 h, both $\lambda$ and $\lambda_{Area}$ are statistically significantly larger in SST4 compared to in CNTL, however the increases are small in magnitude. The fractional area of ascent ($a_u$) is less than 0.5 at all offset times in both CNTL and SST4 (Table 3) indicating that regions of ascent are smaller relative to areas of descent. Warming does not cause large changes to the fractional area of ascent; a statistically significant decrease occurs at $t = 0$ h, a statistically significant increase occurs at $t = +48$ h and no significant changes occur at $t =$-48 h, -24 h or +24 h. Thus the small changes in both asymmetry parameters occur due to changes in the ratio of mean descent to mean ascent. Small increases in the maximum ascent (minimum $\omega$, Table 2) are found and the mean vertical motion averaged over the 12 degree cyclone area also becomes more negative in SST4 (Table 3), indicating either stronger ascent or weaker descent. The mean ascent, calculated over areas where $\omega < 0$ in the 12 degree cyclone area, strengthens slightly at $t =$-24 h, $t =$0 h and $t =$+24 h (Table 3) while the mean descent, averaged over areas where $\omega > 0$ in the 12 degree cyclone area, weakens slightly at all offset times. The magnitude of ratio between the mean upward velocity and mean downward velocity (i.e $|\overline{\omega_u}/\overline{\omega_d}|$) increases at all times, as does the ratio between the mean vertical motion and mean upward velocity (i.e $|\overline{\omega}/\overline{\omega_u}|$), such that $\lambda$ increases with warming.

The magnitude of the mean increase in both $\lambda$ and $\lambda_{Area}$ is small, for example, at the time of maximum intensity $\lambda$ increases from 0.74 in CNTL to 0.77 in SST4 and $\lambda_{Area}$ increases from 0.64 in CNTL to 0.66 in SST4 (Table 3). This result differs slightly from the results of O'Gorman (2011) who, in idealised climate change simulations performed on an aqua-planet, found that $\lambda$ has a value of $\sim$0.6 and that it does not increase when the global mean surface temperature increases. Tamarin-Brodsky and Hadas (2019), however find a small increase in $\lambda$ with warming and no changes to the fractional area of ascent ($a_u$) which is in agreement with what we find here. Previously Booth et al. (2015) calculated $\lambda$, $\lambda_{Area}$ and $a_u$ in dry and moist baroclinic life cycle experiments and find that including moisture increases $\lambda$ from 0.58 to 0.74, increases $\lambda_{Area}$ from 0.55 to 0.64 and decreases $a_u$ from 0.45 to 0.40. These changes are much more pronounced that those found in this study likely due to the relative difference in the two set of experiments, e.g a dry versus moist case compared to a moist case vs a moist case with 4 K warming.

## 8 Conclusions

Aquaplanet simulations were performed with a state-of-the-art, full complexity atmospheric model (OpenIFS) to quantify how the number, characteristics, and structure of extra-tropical cyclones respond to horizontally uniform warming and to identify possible physical reasons for such changes. This simplified "climate change" experimental method was selected because it provides a very large sample size of cyclones for drawing statistically significant conclusions from and because the initial conditions and experimental design do not exert a strong control on the evolution of the model state.

The aquaplanet model set up is capable of producing a zonal mean climate that is broadly similar to that observed on Earth. The response of the zonal mean temperature and zonal mean zonal wind to warming is in broad agreement with multi-model mean predictions from CMIP5 models. Namely, the greatest warming is observed in the tropical upper troposphere, the sub-tropical jet streams intensify, move upwards and polewards with warming, and the eddy-driven jet and mid-latitude storm track moves polewards with warming. The magnitude of the near surface warming in the aqua-planet SST4 simulation compared to CNTL is approximately 4 K which is within the CMIP5 multi-model range predicted to occur by 2100 under the RCP8.5 scenario.

Extra-tropical cyclones were tracked using an objective tracking algorithm which identifies localised maxima of 850-hPa relative vorticity truncated to T42 spectral resolution. In both the control (CNTL) and warm (SST4) experiment about 3500 cyclone tracks were identified. Warming the SSTs did not change the cyclone life time and lead to a 3.3% decrease in the total number of extra-tropical cyclones. Moreover, the median intensity of cyclones, as measured by the maximum 850-hPa vorticity, does not change significantly when SSTs are warmed uniformly, however, the intensity distribution of extra-tropical cyclones broadens resulting in more intense, and more weaker cyclones. The median deepening rate of all extra-tropical cyclones did not change significantly with warming although the zonal mean Eady growth decreased by 5-10% due to an increase in the hydrostatic stability. This apparent conflict arises as moisture acts to intensify the extra-tropical cyclones in these simulations whereas the Eady growth rate is a measure of dry baroclinicity. In addition, both extra-tropical cyclone genesis and lysis regions move poleward with warming which is consistent with the poleward shift of the eddy-driven jet.

These results can be compared to those from previous idealised studies as well as to results obtained from full complexity climate models. Our result that the maximum relative vorticity of the most extreme cyclones increases with warming is in agreement with results from previous aqua-planet simulations (Pfahl et al., 2015). When our results of extra-tropical cyclone intensity are compared to results based on coupled climate models a complex picture emerges. Our result that the number of extreme cyclones increases with warming agrees with the results from Champion et al. (2011) and the southern hemisphere results of Chang et al. (2012) yet disagrees with the results from Bengtsson et al. (2009) and Catto et al. (2011) who both found that the number of intense storms in Europe and the North Atlantic is likely to decrease in the future.

Cyclone composites of the 200 strongest extra-tropical cyclones were created for both the CNTL and the SST4 experiment. The structure of both composite cyclones is qualitatively and even quantitatively very similar to composite cyclones created from reanalysis (Dacre et al., 2012) and historical climate model simulations (Catto et al., 2010). This strongly highlights the validity and usefulness of aqua-planet simulations. The aim of our composite analysis was to identify how the structure of the most intense extra-tropical cyclones responds to warming. The main focus was on how precipitation and vertical motion respond to warming and the omega equation was utilised to assess changes to vertical motion forced by thermal and vorticity advection and attributable to diabatic heating. The main results of how the structure of the 200 most intense extra-tropical cyclones change with warming include:

1. An increase in total column water vapour (TCWV) everywhere within a 12 degree radius of the cyclone centre and an increase in the meridional TCWV gradient. The largest absolute increases in TCWV occur in the warm sector whereas the largest relative increases occur poleward of the warm front.

2. An increase in the 900–700 hPa layer average potential vorticity at all stages of the cyclone evolution everywhere within a 12 degree radius of the cyclone centre. The small absolute increases away from the cyclone centre result from increasing stratification.

3. An increase in the 900-hPa wind speed particularly in the warm sector and thus an increase in the size of the area exposed to wind speeds above a certain fixed threshold.

4. An increase in low-level potential vorticity, total and large-scale precipitation, and ascent at 700 hPa ahead of the warm front at all times of the cyclone life cycle which occurs due to an increase in ascent forced by thermal advection and an increase in how diabatic processes enhance ascent.

5. An expansion of the area of ascent downstream of the cyclone centre due to increased ascent forced by vorticity advection. This is related to a downstream shift and broadening of the 500-hPa trough.

6. A small decrease in the maximum values of ascent at 700 hPa due to vorticity and thermal advection during the cyclone intensification phase and a small increase in the maximum value of ascent due to diabatic heating.

7. A small increase in the asymmetry of vertical motion ($\lambda$ and $\lambda_{Area}$) with warming and no notably changes to the fractional area of ascent. The small increases in asymmetry arises as absolute magnitude of the increase in ascent is greater than the absolute decrease in the magnitude of the descent.

First, these results show that ascent becomes slightly more diabatically driven in the warmer experiment compared to the CNTL experiment and that cyclone-related precipitation increases by up to 50% - a value much larger than predicted for global precipitation amounts. Second, these results indicate that the spatial structure of the most intense extra-tropical cyclones does change with warming. The localised maxima of low-level PV, 900-hPa wind speed, maximum precipitation and vertical velocity associated with the warm front all move north-east relative to the centre of the cyclone. This demonstrates that in the warmer experiment the warm front is farther poleward and downstream of the cyclone centre than in CNTL. Furthermore, the area of ascent also increased with warming particularly in the downstream region due to changes in ascent forced by vorticity advection and ultimately a broader trough at 500 hPa in SST4 compared to in the CNTL experiment.

The cyclone composite analysis revealed that precipitation increased everywhere relative to the cyclone centre with warming. The same result was obtained by Yettella and Kay (2017) who analysed a 30-member initial condition climate model ensemble. Furthermore, Yettella and Kay (2017) find that precipitation in a cyclone composite of Northern hemisphere wintertime extra-tropical cyclones increases from maximum values of ∼9 mm per day to 11 mm per day in the far-future (2081-2100) simulations forced with RCP 8.5. The relative increase is thus smaller than that found in our aqua-planet simulations which is likely due to that in the real world evaporation over land is limited whereas there is always a limitless source of moisture at the surface in an aqua-planet.

The most striking similarity found between our results and previous studies is the downstream shift in the low-level vorticity anomaly and precipitation relative to the cyclone centre. Kirshbaum et al. (2018) and Tierney et al. (2018) both find very similar results of how extra-tropical cyclone structure responds to warming in their baroclinic life cycle experiments. Kirshbaum et al. (2018) show that with increasing environmental temperature the cyclonic potential vorticity associated with the warm front strengthened and moved downstream while Tierney et al. (2018) show that in warmer simulations the upper level PV anomaly is much farther west relative to the low-level PV anomaly than in colder and drier simulations. Thus, this spatial change appears to be a robust feature of how extra-tropical cyclones respond to warming. We thus speculate that in a warmer climate the classical coupling and mutual intensification of lower and upper level anomalies may be disrupted and that extra-tropical cyclone dynamics and associated weather may be notably different.

This study prioritised in-depth understanding of changes to the dynamics and structure of extra-tropical cyclones with warming, rather than quantifying extra-tropical cyclone structure in specific future climate scenarios. Thus, the simulations included numerous simplifications and consequently there are some caveats to this study. First, the aqua-planet simulations contain no polar amplification and thus the low-level temperature gradient does not change with warming. Chang et al. (2012) show that CMIP5 models predict a significant increase in the frequency of extreme cyclones during the winter in the Southern Hemisphere, a result which is in general agreement with our aqua-planet results. This means that (1) our results may be more applicable to the Southern Hemisphere and (2) our results, together with further additional simulations, could be used to ascertain the impact of polar amplification on extra-tropical cyclone intensity. Second, as there is no land in our simulations the

potential impact of differential changes in land-sea temperatures on extra-tropical cyclone dynamics is not considered meaning that our results are likely more applicable to oceanic extra-tropical cyclones. Third, we applied uniform SST warming which neglects localised oceanic cooling that occurs in the northern North Atlantic and parts of the high latitude Southern Ocean in several CMIP5 models (Figure 12.9, Collins et al., 2013). Fourth, the simulations were performed at a resolution more typical of a climate model (125 km) than of a numerical weather prediction model meaning that in both the CNTL and SST4 experiment extra-tropical cyclones may be weaker and precipitation areas broader than if the simulations had been conducted at higher resolution. Nevertheless it appears unlikely that repeating the experiments at higher resolution would fundamentally change the main conclusions as Jung et al. (2012) find that increasing the resolution of the IFS from T159 to T1279 only increases the amount of precipitation by 6% and does not alter the ratio of convective to large-scale precipitation. Lastly, cyclone composites were created from the strongest 200 storms and clearly the results would differ if we considered a larger or smaller number.

We conclude by noting that the results obtained here can be used a stepping stone to better understand predictions from coupled climate models of how the structure of extra-tropical cyclones are likely to change in the future. A logical next step would be to analyse climate model projections for evidence of the downstream shift relative to the cyclone centre of increased low-level potential vorticity, vertical velocity and precipitation.

*Code availability.* OpenIFS is available under license from the European Centre for Medium Range Weather Forecasting (ECMWF). See https://confluence.ecmwf.int/display/OIFS for more details. Information on how to obtain the cyclone identification and tracking algorithm (TRACK) can be found from http://www.nerc-essc.ac.uk/~kih/TRACK/Track.html. The version of the omega equation code applied here is available from Mika Rantanen on request.

*Author contributions.* V. A. Sinclair designed and performed the numerical experiments, analysed the data and wrote the paper. M. Rantanen performed the omega equation calculations. P. Haapanala performed the cyclone tracking and ran the cyclone composite code. J. Räisänän and H. Jarvinen provided guidance on interpreting the results. All authors commented on the manuscript.

*Competing interests.* The authors declare that no competing interests are present.

*Acknowledgements.* We acknowledge ECMWF for making the OpenIFS model available and CSC – IT Center for Science Ltd. for the allocation of computational resources. We thank Glenn Carver, Filip Váňa and Gabriela Szépszó for assistance with OpenIFS and for creating the initial conditions for the simulations. We also thank Kevin Hodges for providing the cyclone tracking code, TRACK, and Helen Dacre for providing the cyclone composite code. VAS is funded by the Academy of Finland (project no. 307331).

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

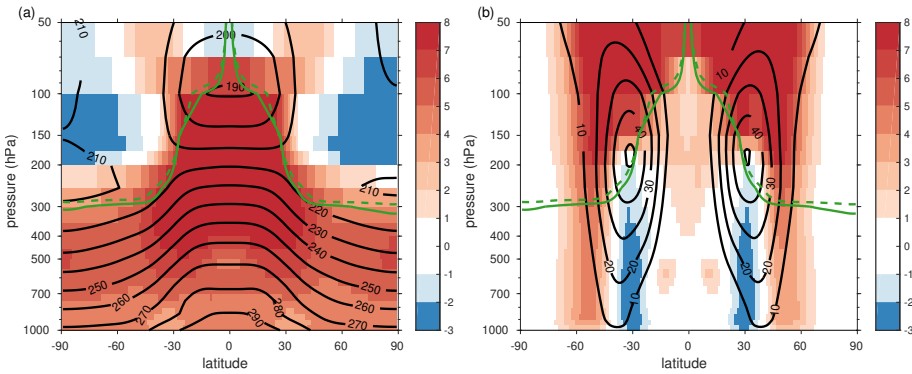

**Figure 1.** Zonal mean (a) temperature (K) and (b) zonal winds (ms$^{-1}$) averaged over 10 years of simulation. Black contours show the CNTL simulation and shading the difference between the SST4 and control simulation (SST4-CNTL). The green solid line shows the dynamic tropopause (the 2 PVU surface) in CNTL and the dashed line in SST4.

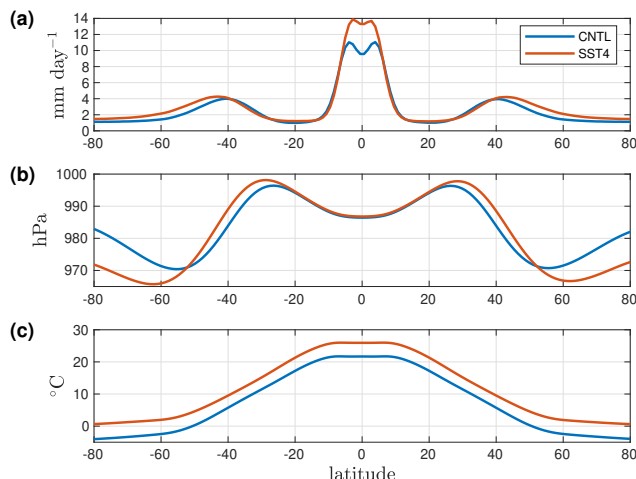

**Figure 2.** Zonal mean (a) total precipitation (mm day$^{-1}$), (b) mean sea level pressure (hPa) and (c) 950-hPa temperature (°C) averaged over 10 years of simulation. Blue line shows CNTL and red SST4.

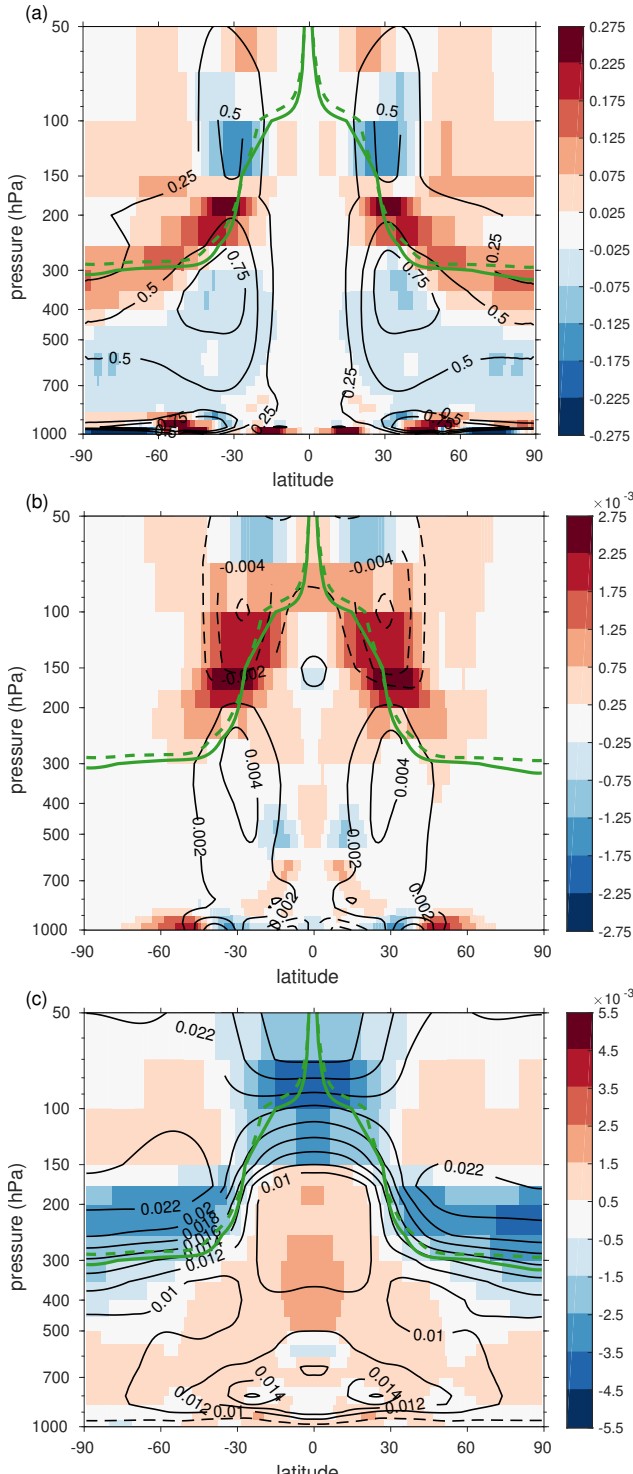

**Figure 3.** Zonal mean (a) maximum Eady growth rate (day$^{-1}$), (b) vertical shear of the zonal wind (s$^{-1}$) and (c) Brunt Väisälä frequency (s$^{-1}$) averaged over 10 years of simulation. Black contours show the CNTL simulation and shading the difference between the SST4 and control simulation (SST4-CNTL). In (c) contours are every 0.002s$^{-1}$ for values greater than 0.01s$^{-1}$ and the dashed line shows the 0.005s$^{-1}$ contour. The green solid line shows the dynamic tropopause (the 2 PVU surface) in CNTL and the green dashed line in SST4.

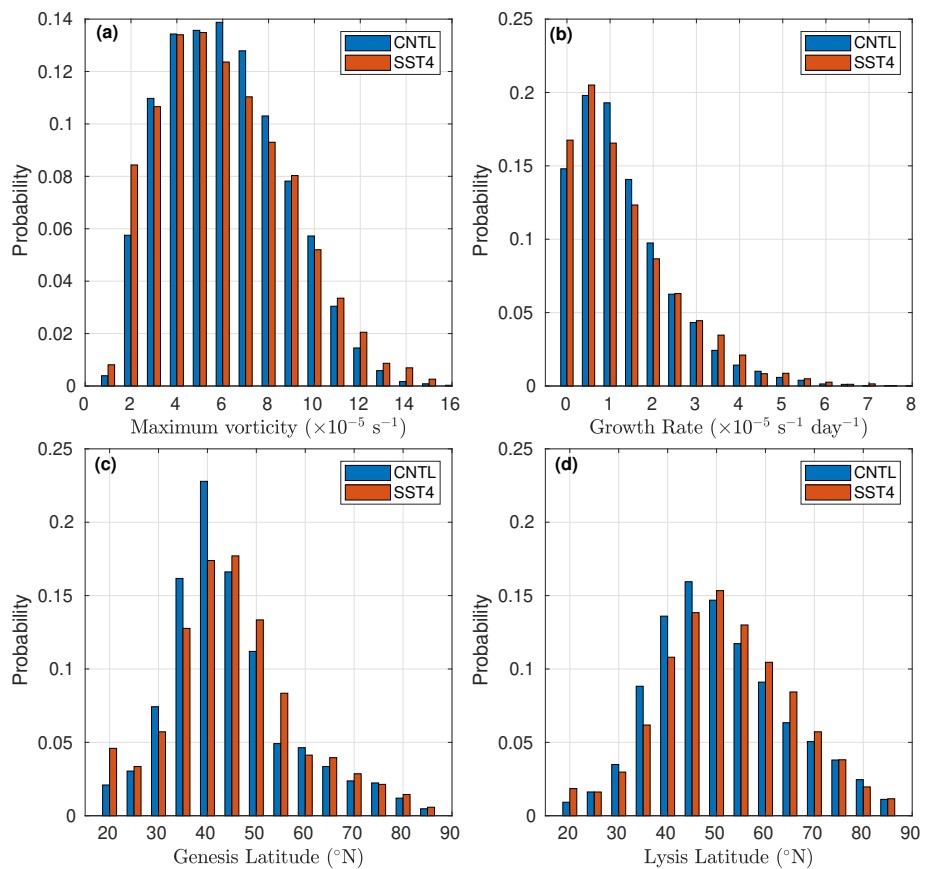

**Figure 4.** Normalized histograms of the extra-tropical cyclone's (a) maximum 850-hPa relative vorticity, (b) average deepening rate between time of genesis and time of maximum vorticity, (c) genesis latitude and (d) lysis latitude. Blue colors show CNTL and red SST4.

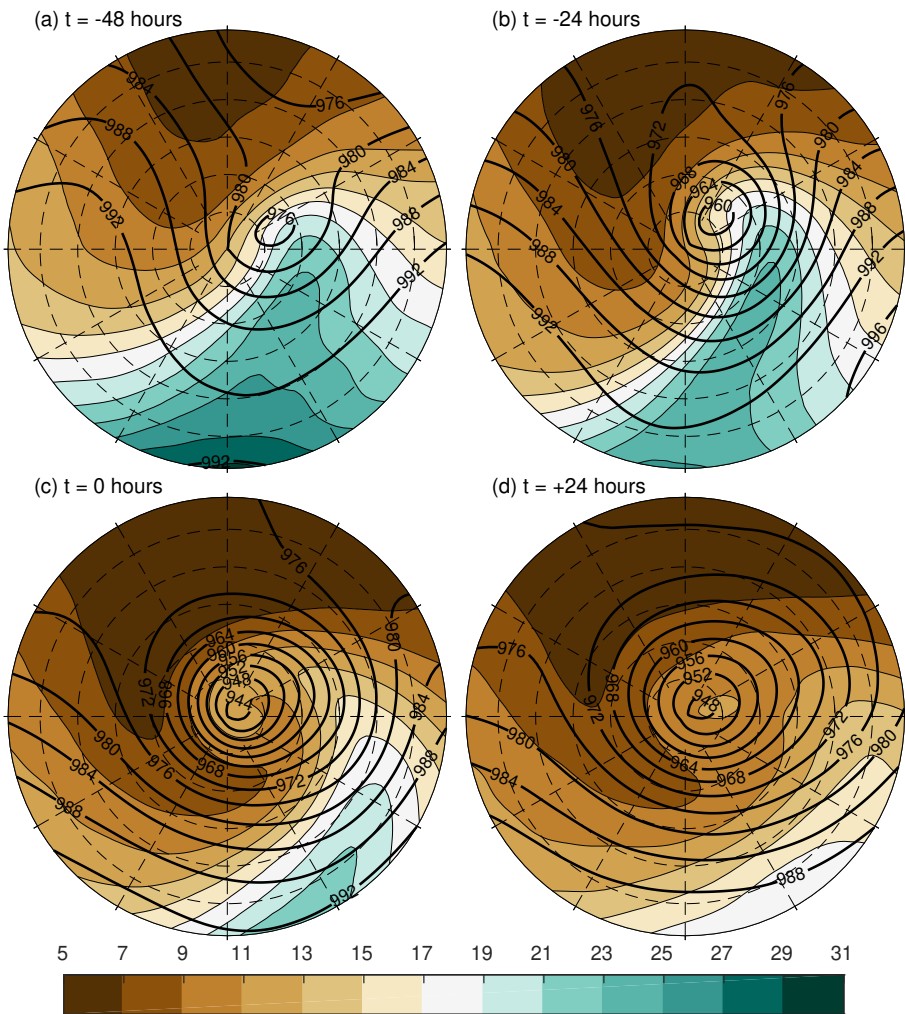

**Figure 5.** Composite cyclone of the strongest 200 extra-tropical cyclones in the CNTL simulation at (a) 48 hours before time of maximum vorticity, (b) 24 hours before time of maximum vorticity, (c) time of maximum vorticity and (d) 24 hours after the time of maximum vorticity. Shading shows the total column water vapour (g kg$^{-1}$) and black contours show the mean sea level pressure (hPa). The plotted radius is 12 degrees.

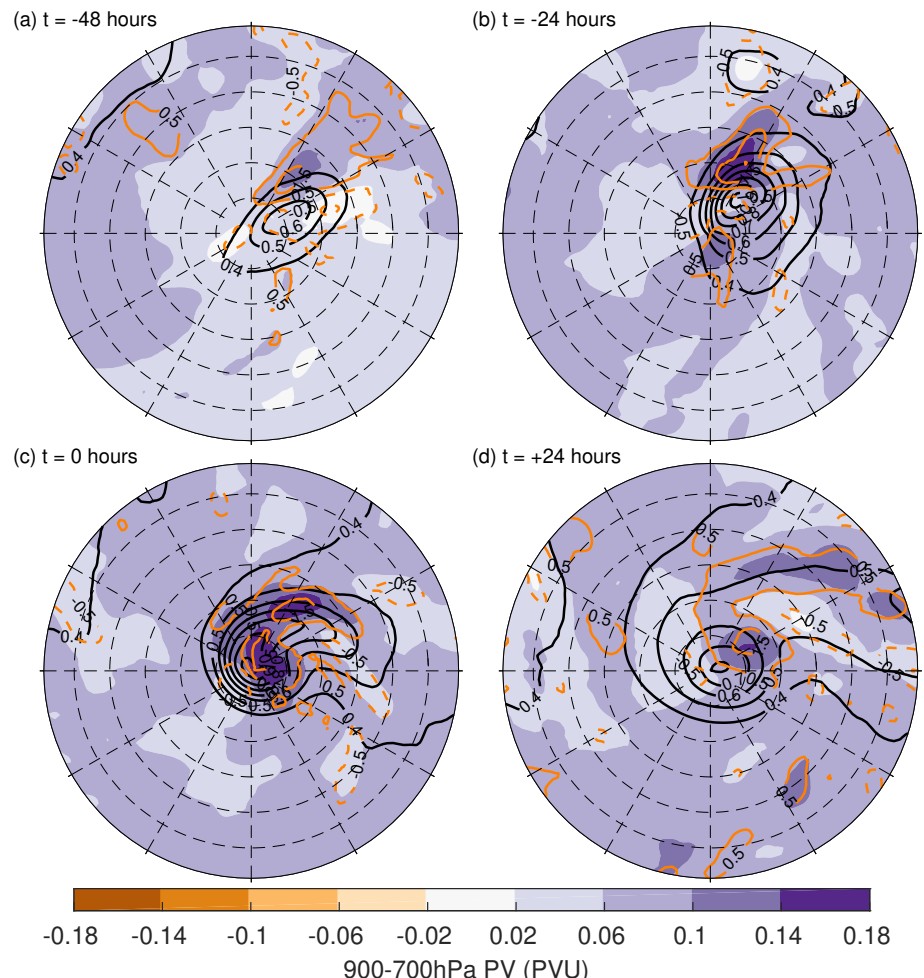

**Figure 6.** Composite mean of the strongest 200 extra-tropical cyclones at (a) 48 hours before time of maximum vorticity, (b) 24 hours before time of maximum vorticity, (c) time of maximum vorticity and (d) 24 hours after the time of maximum vorticity. Black contours show the 900–700 hPa layer mean potential vorticity in CNTL (contour interval 0.1 PVU, starting at 0.4 PVU). Shading shows the difference in the 900–700 hPa layer mean potential vorticity between SST4 and CNTL. Orange contours show the difference in the 850 hPa relative vorticity between SST4 and CNTL (contour interval 0.5 $\times 10^{-5}$ s$^{-1}$, the 0 contour is omitted). Solid orange contours show positive differences and dashed contours negative differences.)

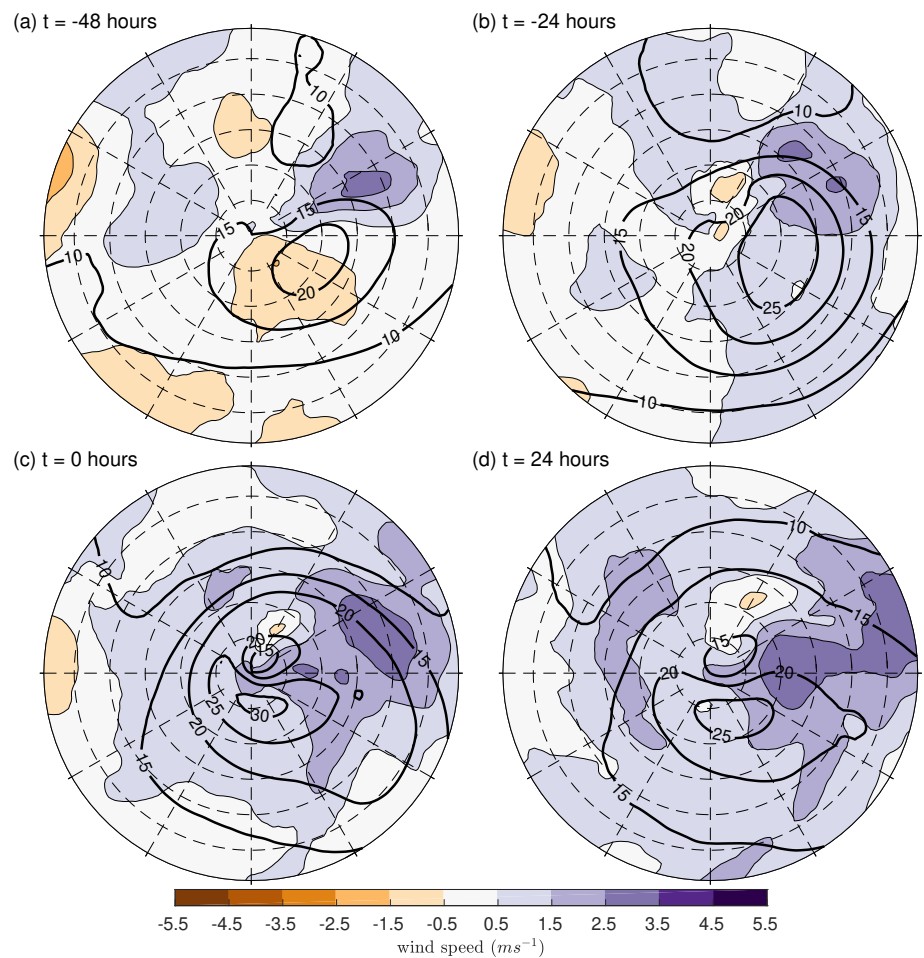

**Figure 7.** Composite mean of the 900-hPa wind speed of the strongest 200 extra-tropical cyclones in the CNTL simulation (black contours, every 5 ms$^{-1}$) and the difference between SST4 and CNTL (shading) at (a) 48 hours before time of maximum intensity, (b) 24 hours before time of maximum intensity, (c) time of maximum intensity and (d) 24 hours after the time of maximum intensity.

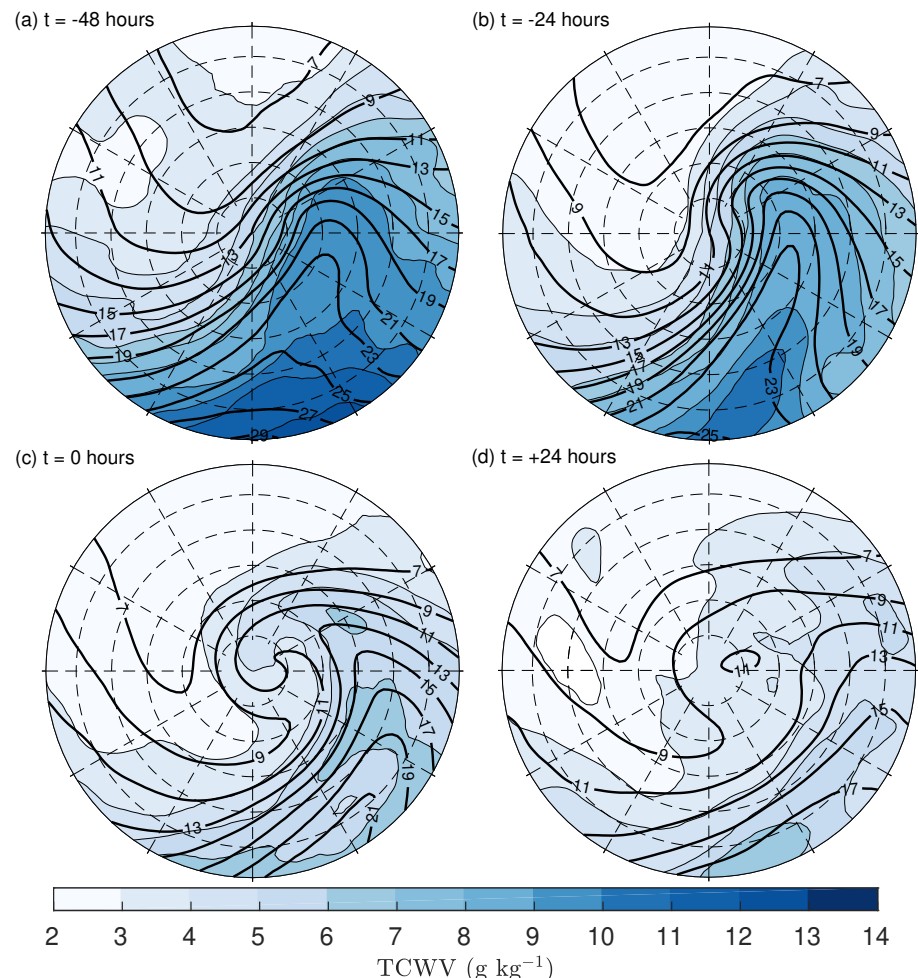

**Figure 8.** Composite mean of the total column water vapour (TCWV) of the strongest 200 extra-tropical cyclones in the CNTL simulation (black contours, every 2 g kg$^{-1}$) and the difference between SST4 and CNTL (shading) at (a) 48 hours before time of maximum intensity, (b) 24 hours before time of maximum intensity, (c) time of maximum intensity and (d) 24 hours after the time of maximum intensity.

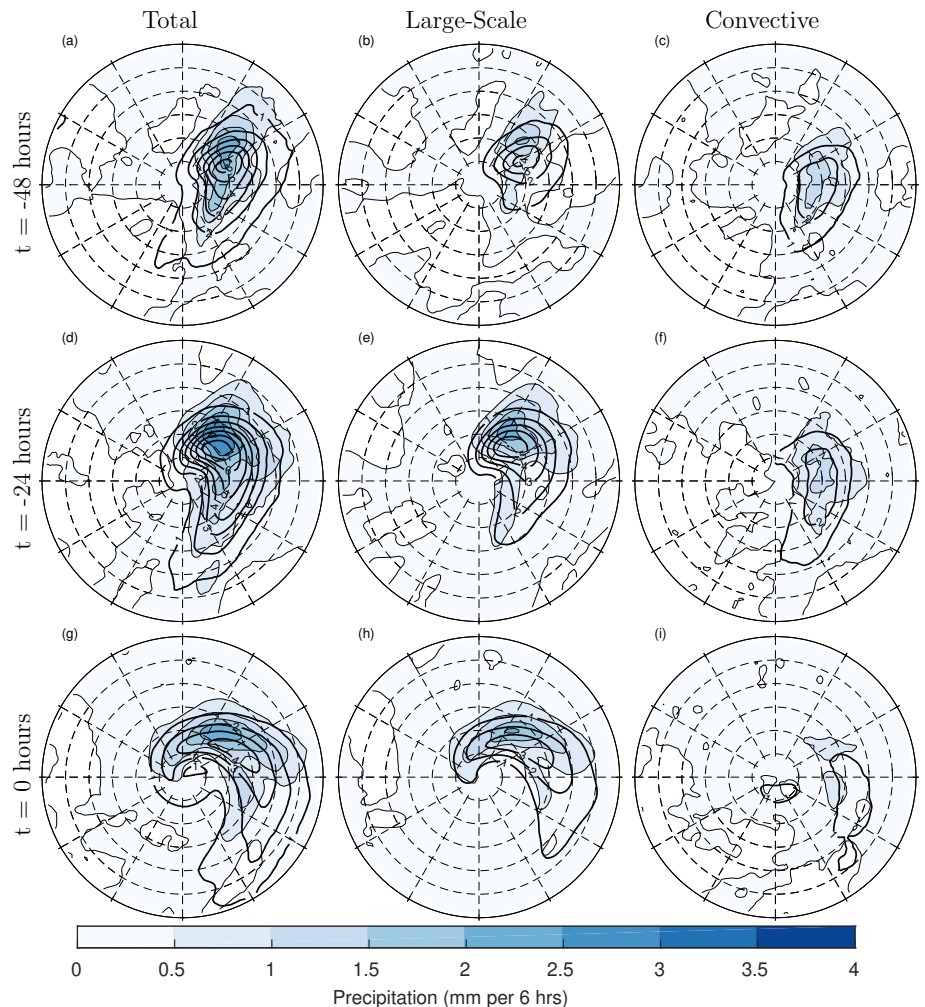

**Figure 9.** Composites of total precipitation (a,d,g), large-scale precipitation (b,e,h) and convective precipitation (c,f,i) in the CNTL simulation (black contours) and the difference between SST4 and control (shading). Panels a-c are valid 48 hours before the time of maximum intensity, panels d-f are valid 24 hours before the time of maximum intensity and panels g-i are valid at the time of maximum intensity. All composites are of the strongest 200 extra-tropical cyclones in each experiment.

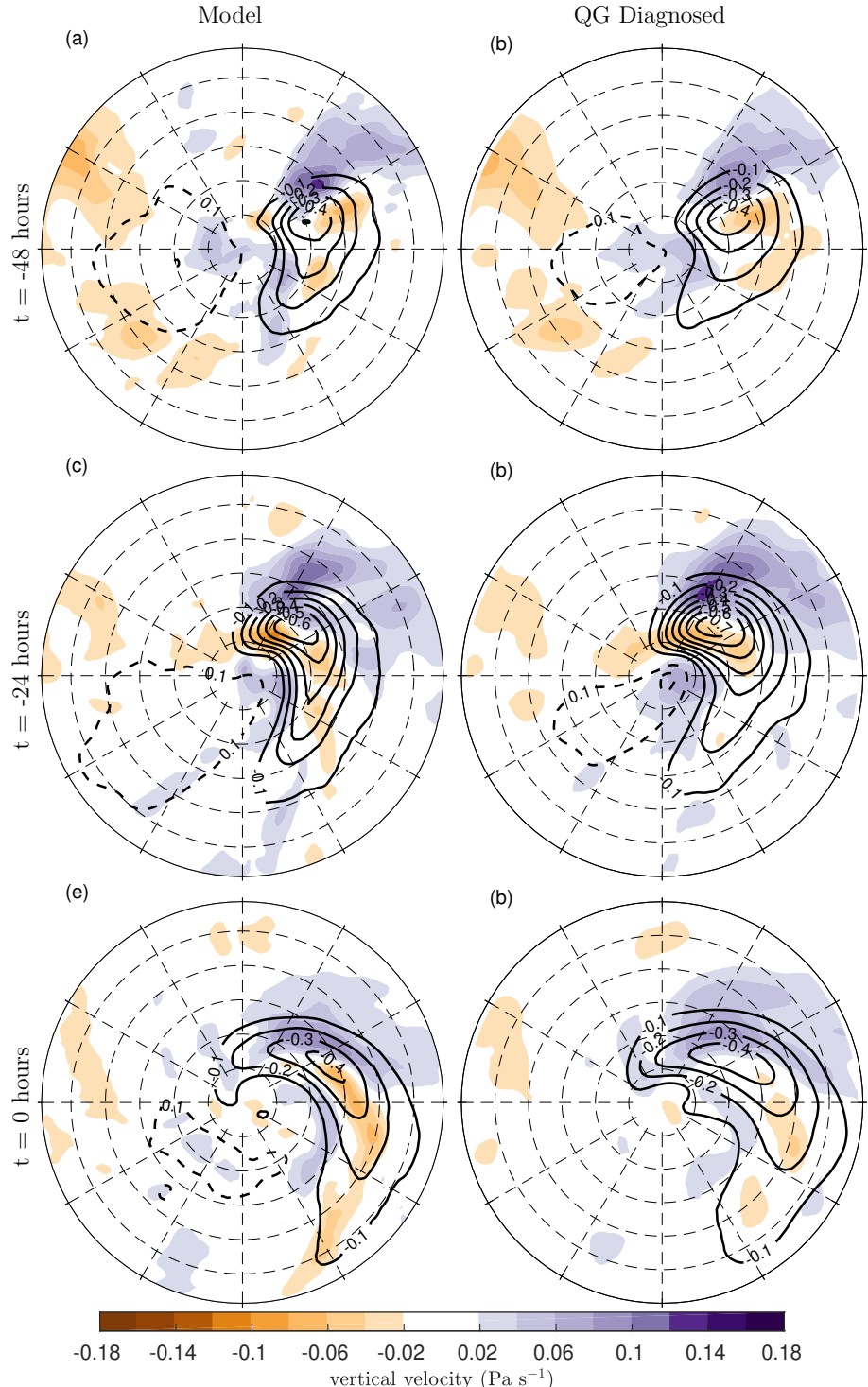

**Figure 10.** Composite mean and change in 700-hPa vertical velocity in pressure coordinates for (a,c,e) total model calculated vertical velocity and (b,d,f) vertical velocity calculated from the modified quasi-geostrophic omega equation (Eq. (1)). Panels a-b are valid 48 hours before the time of maximum intensity, panels c-d are valid 24 hours before the time of maximum intensity and panels e-f are valid at the time of maximum intensity. All composites are of the strongest 200 extra-tropical cyclones in each experiment.

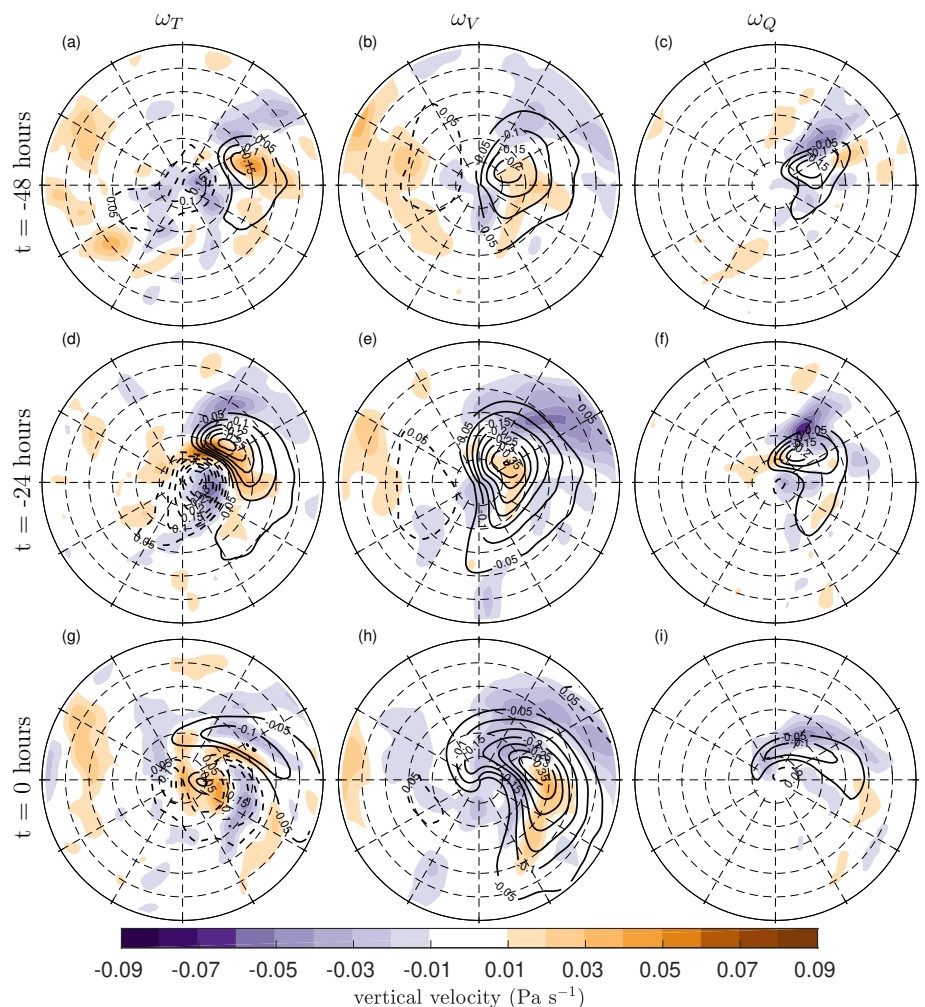

**Figure 11.** Composite mean and change in 700-hPa vertical velocity in pressure coordinates due to thermal advection ($\omega_T$, panels a,d,g), due to vorticity advection ($\omega_V$, panels b,e,h) and due to diabatic heating ($\omega_Q$, panels c,f,i). Panels a-c are valid 48 hours before the time of maximum intensity, panels d-f are valid 24 hours before the time of maximum intensity and panels g-i are valid at the time of maximum intensity. Contours show the control values and shading the difference (SST4 - CNTL). All composites are of the strongest 200 extra-tropical cyclones in each experiment.

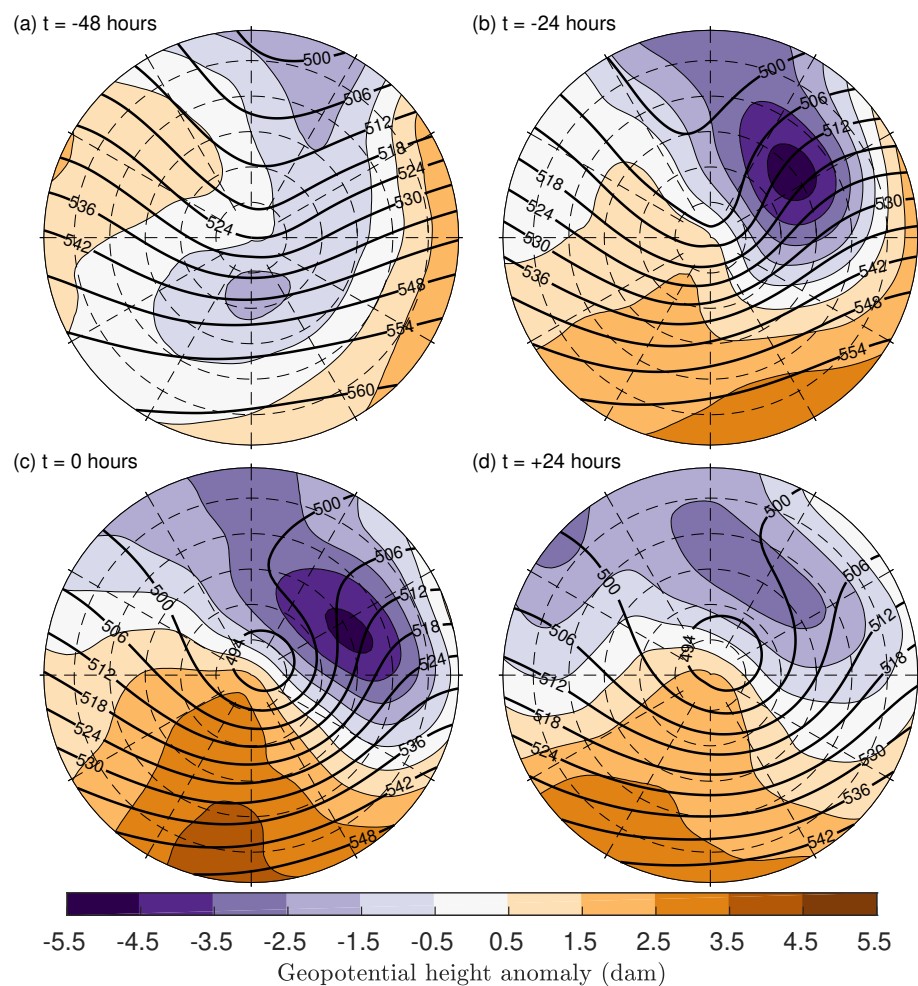

**Figure 12.** Composite mean of 500 hPa geopotential height (dam) in CNTL (black contours) and the difference (SST4-CNTL) in the cyclone relative anomalies at (a) 48 hours before time of maximum intensity, (b) 24 hours before time of maximum intensity, (c) time of maximum intensity and (d) 24 hours after the time of maximum intensity.

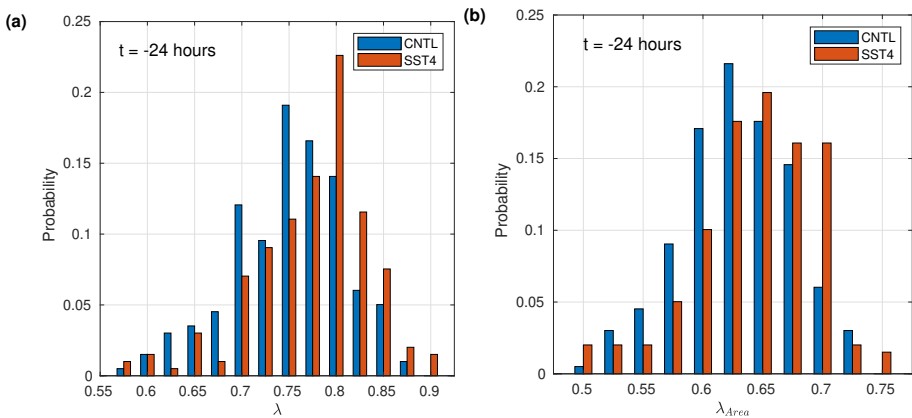

**Figure 13.** Normalized histograms of the (a) asymmetry parameter and (b) area asymmetry parameter (see text for more details) valid 24 hours before the time of maximum intensity. Both histograms include values for the 200 strongest extra-tropical cyclones in both experiments. Blue colors show CNTL and red SST4.

**Table 1.** Cyclone statistics from CNTL and SST4. Relative vorticity values have units of $\times 10^{-5} \text{s}^{-1}$. Duration (life time) is given in units of hours. Deepening rates (units of $\times 10^{-5} \text{s}^{-1} (24\,\text{h})^{-1}$) are the temporal rate of change of the 850-hPa relative vorticity. Positive deepening rate values indicate a strengthening, or deepening, of the extra-tropical cyclone. For vorticity, duration, and deepening rates change is the relative change ((SST4-CNTL)/CNTL) given as a percentage. For genesis and lysis latitude, change is the absolute change.

| Diagnostic | All Cyclones | | | Strongest 200 Cyclones | | |
|---|---|---|---|---|---|---|
| | CNTL | SST4 | Change | CNTL | SST4 | Change |
| Number of tracks / cyclones | 3581 | 3462 | -3.3% | 200 | 200 | 0% |
| Mean maximum 850-hPa vorticity | 6.11 | 6.07 | -0.7% | 11.55 | 11.87 | +2.8% |
| Median maximum 850-hPa vorticity | 5.94 | 5.75 | -3.2% | 11.24 | 11.56 | +2.8% |
| Standard deviation of maximum 850-hPa vorticity | 2.55 | 2.80 | +9.8% | 1.00 | 1.22 | +22% |
| Mean track duration | 132.3 | 127.8 | -3.4% | 209.7 | 190.8 | -9.9% |
| Median track duration | 108.0 | 108.1 | 0% | 192.0 | 180.0 | -6.25% |
| Standard deviation of track duration | 77.3 | 73.5 | -4.9% | 83.0 | 83.2 | +0.24% |
| Median deepening rate (genesis to t = 0 h) | 1.31 | 1.28 | -2.3% | 2.63 | 3.36 | +27.7% |
| Standard deviation of deepening rate (genesis to t = 0 h) | 1.13 | 1.25 | +10.6% | 1.43 | 1.65 | +15.4% |
| Median deepening rate (-24 h to t = 0 h) | 1.42 | 1.43 | +0.7% | 3.57 | 4.41 | +23.5% |
| Standard deviation of deepening rate (-24 h to t = 0 h) | 1.25 | 1.42 | +13.6% | 1.53 | 1.75 | 14.37% |
| median genesis latitude | 44.2°N | 46.2°N | 2.0° | 37.8°N | 38.2°N | +0.4° |
| median lysis latitude | 51.4°N | 53.3°N | 1.9° | 51.2°N | 55.0°N | +3.8° |
| standard deviation of genesis latitude | 12.8° | 13.7° | 0.9° | 8.6° | 8.9° | +0.3° |
| standard deviation of lysis latitude | 13.8° | 14.7° | 0.9° | 11.3° | 11.0° | -0.3° |
| median dlat (lysis - genesis latitude) | 6.2° | 6.0° | -0.2° | 13.7° | 16.7° | +3.0° |
| median dlat (max vort lat -genesis latitude) | 2.9° | 2.9° | 0° | 9.0° | 9.3° | +0.3° |
| 850-hPa relative vorticity threshold for strongest 200 cyclones | - | - | - | 10.44 | 10.88 | +4.2% |
| Vorticity of the strongest cyclone | - | - | - | 15.55 | 16.80 | +8.1% |
| Maximum deepening (genesis to time of max) | - | - | - | 7.05 | 9.10 | +29.0% |

**Table 2.** Maximum values of ascent (Pa s$^{-1}$) at 700 hPa directly from the model ($\omega$) and attributed to vorticity advection ($\omega_V$), thermal advection ($\omega_T$) and diabatic heating ($\omega_Q$) in the CNTL composite mean and the SST4 composite mean at different offset times.

| time (h) | CNTL | | | | SST4 | | | |
|---|---|---|---|---|---|---|---|---|
| | $\omega$ | $\omega_V$ | $\omega_T$ | $\omega_Q$ | $\omega$ | $\omega_V$ | $\omega_T$ | $\omega_Q$ |
| -48 | -0.5016 | -0.2279 | -0.1808 | -0.195 | -0.5238 | -0.2162 | -0.1423 | -0.193 |
| -24 | -0.6722 | -0.401 | -0.3179 | -0.2311 | -0.6965 | -0.3816 | -0.3013 | -0.2339 |
| 0 | -0.4387 | -0.3889 | -0.1517 | -0.1352 | -0.4565 | -0.3761 | -0.1681 | -0.1512 |
| 24 | -0.1572 | -0.1701 | -0.0601 | -0.0371 | -0.1674 | -0.1851 | -0.054 | -0.0358 |

**Table 3.** Mean values of $\lambda$, $\lambda_{Area}$, (see text for definitions), fractional area of ascent ($a_u$), mean vertical motion ($\overline{\omega}$), mean ascent ($\overline{\omega_u}$) and mean descent ($\overline{\omega_d}$) at 700 hPa averaged over the strongest 200 extra-tropical cyclones in CNTL and in SST4 at different offset times.

| time (h) | CNTL | | | | | | SST4 | | | | | |
|---|---|---|---|---|---|---|---|---|---|---|---|---|
| | $\lambda_{Area}$ | $\lambda$ | $a_u$ | $\overline{\omega}$ | $\overline{\omega_u}$ | $\overline{\omega_d}$ | $\lambda_{Area}$ | $\lambda$ | $a_u$ | $\overline{\omega}$ | $\overline{\omega_u}$ | $\overline{\omega_d}$ |
| -48 | 0.60 | 0.70 | 0.30 | 0.0011 | -0.1618 | 0.1083 | 0.59 | 0.70 | 0.30 | 0.0006 | -0.1578 | 0.1065 |
| -24 | 0.63 | 0.75 | 0.41 | -0.0137 | -0.1922 | 0.1135 | 0.64 | 0.77 | 0.40 | -0.0178 | -0.1974 | 0.1091 |
| 0 | 0.64 | 0.74 | 0.43 | -0.0183 | -0.1801 | 0.1021 | 0.66 | 0.77 | 0.42 | -0.0220 | -0.1890 | 0.0966 |
| 24 | 0.60 | 0.69 | 0.45 | -0.0116 | -0.1322 | 0.0867 | 0.62 | 0.73 | 0.47 | -0.0162 | -0.1329 | 0.0805 |
| 48 | 0.59 | 0.69 | 0.41 | -0.0066 | -0.1142 | 0.0777 | 0.60 | 0.71 | 0.42 | -0.0091 | -0.1053 | 0.0703 |