# Peer review of "The characteristics and structure of extra-tropical cyclones in a warmer climate"

_Weather and Climate Dynamics, 2019_

## Referee Comment (RC1) · Anonymous Referee #1 · 29 Sep 2019

This paper uses Lagrangian tracking to explore extratropical cyclones structure in an aquaplanet: for an SST distribution that gives a global climate similar to present day and then a second time with a 4K addition to all SST. The study finds that storms in the modeled warmer climate have a larger contribution to the generation of their dynamical strength from processes associated latent heating. This is a result that has been shown before, but not with this type of model. That being said, this manuscript can still be judged to advance the field, because of the detailed analysis of the cyclone circulation.

Overall, I think the work is well written. I also think it could be improved if the placed their results in the context of existing literature. I also have multiple minor recommendations that I hope will be addressed.

[Figure]

Minor #1: I think the authors need discuss the signal-to-noise issue in greater detail. The signal being the change in storms with warming, and the noise being interannual and intraseasonal variability (as partially discussed in: Harvey et al. GRL 2012).

My sense is that a +4 SST warming represents something like how the midlatitudes will look in 150 years or so, but perhaps that is optimistic? From my eye-ball comparison of your zonal mean changes, with that of Lu et al. 2008 (because I had it on hand, and I did not want to page through the entire report that you cited – Collins et al. 2013) – it looks like your model is warming in the troposphere by 2 - 3 more degrees than theirs. My point being, it would be good to express some idea of what your models warming signal is akin to timing wise – rather than just saying that the zonal mean changes look similar to the CMIP5 models. Or, another way to express my comment is: what time horizon do you expect that we will be able to observe changes like those shown in the paper? I think information like this would make the paper more applicable.

Also relevant: for some of the results, the signal you are finding is in the top 200 events. What do things look like if you use a larger set? I think it is fine to report on the changes in the extremes, but then in the discussion, or elsewhere, I would like you to contextualize this, in terms of your analysis method.

Lu, J., G. Chen, and D.M. Frierson, 2008: Response of the Zonal Mean Atmospheric Circulation to El Niño versus Global Warming. J. Climate, 21, 5835–5851.

Harvey, B. J., Shaffrey, L. C., Woollings, T. J., Zappa, G., and Hodges, K. I. ( 2012), How large are projected 21st century storm track changes?, Geophys. Res. Lett., 39, L18707, doi:10.1029/2012GL052873.

Minor #2: Things that might need to be mentioned in the conclusions, or considered for further study: 1) With global warming projections, it seems likely that differential changes in land-sea temperatures (or local cooling of SST in certain regions) will have important impacts on the dry dynamical forcing of the cyclones. This cannot be captured with an aquaplanet. Thus, perhaps, these results are mainly relevant to cyclones

in the middle of the ocean. Which is fine, but it needs to be mentioned. 2) Climate models are known to have biases in the representation of clouds and precipitation (raining too frequently at weak rain rates and not frequently enough at strong rates), however, they capture the bulk impact of precipitation. So how confident are you in the model results, especially those related to precipitation and vertical velocity.

Relevant to comment #2 and Section 6.5: One step you can take, to provide more context on your model result relative to existing studies, is to examine the asymmetry (e.g., Tamarin-Brodsky and Hadas, 2019, and citations therein), in the vertical motion in the model's current climate and SST+4K climate.

Tamarin-Brodsky T, Hadas O. The Asymmetry of Vertical Velocity in Current and Future Climate. Geophys Res Lett. 2019 Jan 16;46(1):374–82.

Minor #3: Are you sure that you are not finding some tropical cyclones? Your procedure for excluding them does not seem too robust. I know from experience with SLP trackers, that such an approach would not exclude all tropical cyclones. Perhaps it is more straight forward with relative vorticity trackers?

Line-by-line

Line 43: There are also studies of the cloud and precipitation structure taken from satellite retrievals. Given that reanalysis is still reliant on a model for precipitation and cloud physics, I think it is important to keep studies like these on people's minds:

Naud, C. M., A. D. Del Genio, M. Bauer and W. Kovari, 2010: Cloud vertical distribution across warm and cold fronts in CloudSat-CALIPSO data and a general circulation model. J. Climate, 23, 3397-3415.

Naud, C. M., J. F. Booth, Lebsock, M. and M Grecu, 2018: Observational Constraint for Precipitation in Extratropical Cyclones: sensitivity to data sources. Journal of Applied Meteorology and Climatology, 57, 991–1009, https://doi.org/10.1175/JAMC-D-17-0289.1

Line 55: Some other highly relevant paper on this topic. These could be useful both in the introduction and, perhaps, in the conclusions.

Champion AJ, Hodges KI, Bengtsson LO, Keenlyside NS, Esch M (2011) Impact of increasing resolution and a warmer climate on extreme weather from Northern Hemisphere extratropical cyclones. Tellus 63A:893-906

Yettella, V., & Kay, J. E. (2017). How will precipitation change in extratropical cyclones as the planet warms? Insights from a large initial condition climate model ensemble. Climate Dynamics, 49(5-6), 1765–1781. https://doi.org/10.1007/s00382-016-3410-2

Michaelis, A. C., Willison, J., Lackmann, G. M., & Robinson, W. A. (2017). Changes in winter North Atlantic extratropical cyclones in high- resolution regional pseudo–global warming simulations. Journal of Climate, 30(17), 6905–6925. https://doi.org/10.1175/JCLI-D-16-0697.1

Line 185: Can you briefly discuss the level of agreement between the omega that you estimate and that which is calculated by the model? I am curious to know how far off they are. Related to this, in Section 365, are your discussing the model-produced omega or that which you estimate from this equation?

Line 223: From the figure, it is difficult to see what is occurring for low-level baroclinicity (e.g., below 900 hPa), because of the contours. Is it getting stronger or weaker near the surface?

Line 225: Be careful in how about how you state this. I agree that stratospheric baroclinicity is likely to have a small role. But near the tropopause things are less clear, see for instance: Yuval, J. and Y. Kaspi, 2016: Eddy Activity Sensitivity to Changes in the Vertical Structure of Baroclinicity. J. Atmos. Sci., 73, 1709–1726

Line 270: This latitude shift is going to have a big impact on the precipitation (e.g., Booth et al. 2018). It seems a bit confusing that the storm track as a whole shifts polewards, but the strongest events initialize closer to the equator. This likely relates to

the added role of moisture in driving the strength?

Booth, J. F., Naud, C. M., & J. Jeyaratnam, 2018: Extratropical cyclone precipitation life cycles: A satellite-based analysis. Geophysical Research Letters, 45, 8647-8654.

Line 501: Tierney et al. (2018) also documented this shift. My question: can you speculate on what impact this for people, either on the hazards created by the cyclones or the interactions between the cyclone mid-level and upper-level circulation. Can we speculate that in a warmer world the storms mid-level disconnects from the upper-level – which could have a big impact on storm behavior? Isn't this what Tierney et al. and Kirshbaum et al find with their baroclinic wave studies? Or are is this change in the structure of the cyclones in warming simulations just a curiosity that is of interest to the dynamics community? I think it is the former, but things like this should be stated and discussed a bit so if you want the paper to reach outside the cyclones community.
* * *

---

## Referee Comment (RC2) · Anonymous Referee #2 · 7 Oct 2019

This is a very nice study looking at aquaplanet model simulations with a complex model. This provides a step in the model hierarchy between fully complex models (e.g. the CMIP models) and the very idealised models such as a baroclinic channel models. By using a Lagrangian feature tracking algorithm and looking at the lifecycles of extratropical cyclones, the authors have investigated the changing intensity and structure of the 200 most intense cyclones. The paper is well-written and the figures are very clear.

I have a few minor comments and suggestions to make.

1. Line 208: The CMIP5 model projections are not shown here so it would be good to refer to a paper that shows these.

[Figure]

2. Line 213: Is the poleward change of 2.2 degrees significant? This is much smaller than the shift in the jet.

3. Line 219: Is there an explanation for the much lower average MSLP? Is this an issue with OpenIFS or something to do with the aquaplanet set up?

4. Lines 317-321: It would be good to see figures for the changes in the wind speeds. Both here and in the precipitation section, I think a useful addition would be some analysis of the footprints of the most intense winds and precipitation and how this changes. These footprints were considered in the Tierney et al 2018 and Pfahl et al 2015 papers.

5. Line 350: I think this sentence should be reworded slightly – it seems that what is consistent is that the condensation from the precipitation gives more latent heating and stronger PV anomaly (rather than the latent heating leading to more precipitation).

6. Line 357: It would be good to make it clearer here and elsewhere in the paragraph when it is referring to changes in the SST4 experiment.

7. Line 376-377: I think it might be good to say the warm conveyor belts are further poleward relative to the propagation direction rather than the cyclone centre since the cyclones have been rotated for compositing.

8. Line 416: I think the wrong figure panel is referenced here – it should be 10d.

9. Section 7: It would be nice to see greater discussion of this study in the context of previous literature. For example, how do these results compare with, e.g. Pfahl et al 2015, Tierney et al 2018. Are there any other papers that analyse the structure of extratropical cyclones in the future? Two that I can think of are Yettella and Kay 2017 and Michaelis et al 2017.

10. Section 7: Also, are there any caveats with the study – would the results change if you looked at the 500 most intense storms or the medium intensity storms?

References: Yettella V, Kay JE. How will precipitation change in extratropical cyclones as the planet warms? Insights from a large initial condition climate model ensemble. Clim Dyn. 2017;49(5–6):1765–81.

Michaelis AC, Willison J, Lackmann GM, Robinson WA. Changes in Winter North Atlantic Extratropical Cyclones in High-Resolution Regional Pseudo–Global Warming Simulations. J Clim. 2017 Jun 6;30(17):6905–25.
* * *

---

## Editor Comment (EC1) · Sebastian Schemm (Editor) · 7 Oct 2019

(1) Figure 6 shows the difference between SST4 and CNTL for the 900–700 hPa layer mean potential vorticity. There are two maxima, for example in Fig.6c (t=0), the first near the occlusion point (in the northeast sector of the cyclone) and the second near the bent-back front (close to the composite center, corresponding to the vorticity maximum identified by the tracking).

However, the corresponding equivalent for precipiation, Fig. 8g (t=0), shows only one maxima, which is in the north-eastern sector of the cyclone. This suggests, that the potential vorticity seen in Fig.6c in the north-eastern sector of the cyclone is formed by enhanced diabatic processes, while the second potential vorticity anomaly near the

bent-back front is resulting from enhanced advection.

This difference could be explained by the linkage between the cold and the warm conveyor belts. The positive potential vorticity anomaly, which is diabatically generated in the air below the rising warm conveyor belt, is advected by the cold conveyor along the bent-back front, where it contributes to the enhanced potential vorticity gradients (corresponding to higher wind speed near the tail of the bent-back front of the cyclone). When the linkage is accelerated, the first low-level potential vorticity maximum is explained by enhanced diabatic processes and the second by accelerated advection by the cold conveyor belt, resulting in only one maxima in the precipitation pattern but two in the potential vorticity pattern, which is in agreement with the here presented composites (Fig.6 and Fig.8). The linkage between the conveyor belts has been described in an idealized setting in Schemm and Wernli (2014) and is summarized in their Figure 9.

Schemm, S. and H. Wernli, 2014: The Linkage between the Warm and the Cold Conveyor Belts in an Idealized Extratropical Cyclone. J. Atmos. Sci., 71, 1443–1459, https://doi.org/10.1175/JAS-D-13-0177.1 (A video that helps to illuminate the linkage is provided at https://journals.ametsoc.org/doi/suppl/10.1175/JAS-D-13-0177.1)

(2) A couple of suggested additional literature that seems to be in agreement with the findings of the submitted manuscript.

Regarding the changes eddy intensity:

Paul A. O'Gorman 2010: Understanding the varied response of the extratropical storm tracks to climate change. Proceedings of the National Academy of Sciences Nov 2010, 107 (45) 19176-19180; DOI: 10.1073/pnas.1011547107

O'Gorman, P.A. and T. Schneider, 2008: Energy of Midlatitude Transient Eddies in Idealized Simulations of Changed Climates. J. Climate, 21, 5797–5806, https://doi.org/10.1175/2008JCLI2099.1

And in agreement with the fact that the large-scale eddies appear to stabilize the tropopshere in a warmer climate:

Korty, R.L. and T. Schneider, 2007: A Climatology of the Tropospheric Thermal Stratification Using Saturation Potential Vorticity. J. Climate, 20, 5977–5991, https://doi.org/10.1175/2007JCLI1788.1

---

## Author Comment (AC1) · 6 Nov 2019

**Response to Reviewers - "The characteristics and structure of extra-tropical cyclones in a warmer climate"**

Victoria A. Sinclair, Mika Rantanen, Päivi Haapanala, Jouni Räisänen and Heikki Järvinen

November 6, 2019

We thank the reviewer for their constructive comments on our submitted manuscript. We have copied the comments of reviewer 1 in black here and include our response to each individual comment in blue.

**Reviewer 1**

This paper uses Lagrangian tracking to explore extratropical cyclones structure in an aquaplanet: for an SST distribution that gives a global climate similar to present day and then a second time with a 4K addition to all SST. The study finds that storms in the modeled warmer climate have a larger contribution to the generation of their dynamical strength from processes associated latent heating. This is a result that has been shown before, but not with this type of model. That being said, this manuscript can still be judged to advance the field, because of the detailed analysis of the cyclone circulation.Overall, I think the work is well written. I also think it could be improved if the placed their results in the context of existing literature. I also have multiple minor recommendations that I hope will be addressed.

1. Minor 1: I think the authors need discuss the signal-to-noise issue in greater detail. The signal being the change in storms with warming, and the noise being interannual and intraseasonal variability (as partially discussed in: Harvey et al. GRL 2012).

   This is valid point concerning the signal to noise ratio. It should be noted that in these simulations there is no seasonal variation (as we state in section 2.2) which removes some variability in these simulations as does the use of fixed SSTs. Previously we had include the standard deviation of the maximum 850-hPa vorticity, track duration and deepening rate in Table 1. We have now added the standard deviation of genesis and lysis latitude and discuss these values in relation to the absolute change in section 5. We also calculated the inter-annual variability in the mean number of cyclone tracks per year, the mean maximum vorticity, genesis latitude and lysis latitude and discuss these values in relation to the changes between CNTL and SST4 in section 5.

   To assess if the change in cyclone structure is large relative to the inter-annual variability (or variability in cyclones within one experiment) is very difficult as we do not have enough cyclones per year ($\sim$350) to create meaningful composites of the strongest 200 cyclones. We estimated the

variation within the composite cyclone in the CNTL and SST4 experiments by calculating the standard deviation of variables at each spatial point in the composites (not shown). However, the standard deviations are large primarily because the fronts are not in the same place in each cyclone (even though we rotate the cyclones to minimise this issue) and thus this method is not appropriate.

My sense is that a +4 SST warming represents something like how the mid-latitudes will look in 150 years or so, but perhaps that is optimistic? From my eye-ball comparison of your zonal mean changes, with that of Lu et al. 2008 (because I had it on hand, and I did not want to page through the entire report that you cited – Collins et al. 2013) – it looks like your model is warming in the troposphere by 2 - 3 more degrees than theirs. My point being, it would be good to express some idea of what your models warming signal is akin to timing wise – rather than just saying that the zonal mean changes look similar to the CMIP5 models. Or, another way to express my comment is: what time horizon do you expect that we will be able to observe changes like those shown in the paper? I think information like this would make the paper more applicable.

This is a good point. In the 5th Assessment report summary for policy makers, it states "The global mean surface temperature change for the period 2016–2035 relative to 1986–2005 will likely be in the range of 0.3°C to 0.7°C" therefore the changes in our model simulation are not expected in the next 2 decades. However, the AR5 summary report also states that "Increase of global mean surface temperatures for 2081–2100 relative to 1986–2005 is projected to likely be in the ranges derived from the concentration-driven CMIP5 model simulations, that is, 0.3°C to 1.7°C (RCP2.6), 1.1°C to 2.6°C (RCP4.5), 1.4°C to 3.1°C (RCP6.0), 2.6°C to 4.8°C (RCP8.5)." Therefore, in the worst case scenario (RCP8.5), the increase in lower tropospheric temperatures that we see (of order 4K) could be obtained by 2100. We have added relevant references and discussion of this to section 4 and to the conclusions.

Also relevant: for some of the results, the signal you are finding is in the top 200 events. What do things look like if you use a larger set? I think it is fine to report on the changes in the extremes, but then in the discussion, or elsewhere, I would like you to contextualise this, in terms of your analysis method. We did not look at a larger set of cyclones however we expected that for each composite (e.g. CNTL and SST4) the cyclones would be weaker and less structure would be evident as adding more (weaker) cyclones would add to the variability. We add text on this to the conclusions section. We did consider composites of the median cyclones in both CNTL and SST4 and figures of these are now included in the supporting material. The main conclusion is that when the median cyclones are considered the increase in total column water vapour and precipitation are much smaller in absolute terms compared to when the strongest 200 cyclones are considered. Furthermore, the spatial changes to vertical velocity and precipitation are less coherent and differ from what is found for the strongest 200 cyclones.

Lu, J., G. Chen, and D.M. Frierson, 2008: Response of the Zonal Mean Atmospheric Circulation to El Niño versus Global Warming. J. Climate, 21, 5835–5851.

Harvey, B. J., Shaffrey, L. C., Woollings, T. J., Zappa, G., and Hodges, K. I. (2012), How large are projected 21st century storm track changes?, Geophys. Res. Lett., 39,L18707, `doi: 10.1029/2012GL052873`.

2. Minor 2: Things that might need to be mentioned in the conclusions, or considered for further study: 1) With global warming projections, it seems likely that differential changes in land-sea temperatures (or local cooling of SST in certain regions) will have important impacts on the dry dynamical forcing of the cyclones. This cannot be captured with an aquaplanet. Thus, perhaps, these results are mainly relevant to cyclones in the middle of the ocean. Which is fine, but it needs to be mentioned.

We now include a comment about this in the conclusions section alongside the text where we had already mentioned the limitation that an aqua-planet lacks polar amplification.

2) Climate models are known to have biases in the representation of clouds and precipitation (raining too frequently at weak rain rates and not frequently enough at strong rates), however, they capture the bulk impact of precipitation. So how confident are you in the model results, especially those related to precipitation and vertical velocity.

We fully agree that coupled climate models have biases. However, we are using an atmospheric only model with fixed SSTs, and with state-of-the-art microphysics and convection schemes which hopefully limits the bias (although in an idealised simulation it is impossible to assess any model bias). We compared our cyclone composites to those from reanalysis [Dacre et al., 2012] and from satellite based observations and the structure and magnitude of both the vertical velocity and precipitation fields agree quite well. We have added a point about this comparison to the conclusions.

We do agree that our simulations were performed at similar resolution to which climate models are run at (T159, 125 km grid spacing) and we think this coarse resolution is the main source of model error and may results in broader precipitation areas. Jung et al. [2012] compared the impact of the horizontal resolution (from T159 to T1279) of the IFS on precipitation and on the number and location of extra-tropical cyclones. They find that increasing resolution from T159 to T1279 increases average precipitation in the 20-90°N latitude band by 6% but that the ratio of large-scale to convective precipitation does not change. Jung et al. [2012] also find that the intensity of the most extreme extra-tropical cyclones is underestimated at T159 resolution (compared to reanalysis at T255 resolution) and that number of extra-tropical cyclones increases by 6% in winter when resolution is increased from T159 to T1279. Nevertheless, they ultimately conclude that "the influence of increased horizontal resolution in the extra-tropics on the mean atmospheric circulation of the IFS turns out to be relatively small". We have added a discussion about the likely impact of model resolution on our results to the conclusions.

Relevant to comment 2 and Section 6.5: One step you can take, to provide more context on your model result relative to existing studies, is to examine the asymmetry (e.g., Tamarin-Brodsky and Hadas, 2019, and citations therein), in the vertical motion in the model's current climate and SST+4K climate.

Tamarin-Brodsky T, Hadas O. The Asymmetry of Vertical Velocity in Current and Future Climate. Geophys Res Lett. 2019 Jan 16;46(1):374–82.

This is a good suggestion and in response we now include an extra figure, one table and discussion in a new section about how the asymmetry parameter varies between the control and SST4 experiments at different offset times.

3. Minor 3: Are you sure that you are not finding some tropical cyclones? Your procedure for excluding them does not seem too robust. I know from experience with SLP trackers, that such an approach would not exclude all tropical cyclones. Perhaps it is more straight forward with relative vorticity trackers?

We have visually inspected each individual cyclone included in the composites (in total 400 storms) and none of these have characteristics of tropical cyclones. We considered plots of mean sea level pressure and relative vorticity and frontal, asymmetric features were present in all cyclones.

It is harder to establish if we have excluded all tropical cyclones from the results presented in Figure 4 and in Table 1. We considered various different filtering options to attempt to remove tropical cyclones. In the manuscript we remove tracks which did not have at least one point north of 20°N. The genesis and lysis points, coloured by the maximum vorticity of the track are shown in Figure 1 in this response. The tracks which remain at tropical latitudes are weak cyclones and therefore they are unlikely to be tropical cyclones. Another option we considered was to filter based on genesis latitude but this is problematic as it removes systems which undergo extra-tropical transition which we feel should be retained as they do become extra-tropical cyclones. We also attempted filtering on genesis and lysis latitudes but this mainly removed weak cyclones and did not change the results shown in Figure 4. Therefore, given this, and the subjective nature of defining latitude thresholds (particularly given the tropics expands meridionally with warming), we did not make any revisions to the manuscript on this point.

[Figure]

Figure 1: Scatter plots showing genesis latitude versus lysis latitude for all tracks retained for Figure 4 and Table 1. Colour shading shows the maximum vorticity obtained along each track.

**1 Line-by-line**

1. Line 43: There are also studies of the cloud and precipitation structure taken from satellite retrievals. Given that reanalysis is still reliant on a model for precipitation and cloud physics, I think it is important to keep studies like these on people's minds:

Naud, C. M., A. D. Del Genio, M. Bauer and W. Kovari, 2010: Cloud vertical distribution across warm and cold fronts in CloudSat-CALIPSO data and a general circulation model. J. Climate, 23, 3397-3415.

Naud, C. M., J. F. Booth, Lebsock, M. and M Grecu, 2018: Observational Constraint for Precipitation in Extratropical Cyclones: sensitivity to data sources. Journal of Applied Meteorology and Climatology, 57, 991–1009.

Thank you for the suggestion. We now include a few sentences in the introduction about extratropical cyclone studies based on satellite data, including these two references and another paper by Field and Wood (2007).

2. Line 55: Some other highly relevant paper on this topic. These could be useful both in the introduction and, perhaps, in the conclusions.

Champion AJ, Hodges KI, Bengtsson LO, Keenlyside NS, Esch M (2011) Impact of increasing resolution and a warmer climate on extreme weather from Northern Hemisphere extratropical cyclones. Tellus 63A:893-906

Yettella, V., and Kay, J. E. (2017). How will precipitation change in extratropical cyclones as the planet warms? Insights from a large initial condition climate model ensemble. Climate Dynamics, 49(5-6), 1765–1781.

Michaelis, A. C., Willison, J., Lackmann, G. M., and Robinson, W. A. (2017). Changes in winter North Atlantic extratropical cyclones in high- resolution regional pseudo–global warming simulations. Journal of Climate, 30(17), 6905–6925.

Thank you for these suggestions. We now include some in the introduction to motivate the work and others in the conclusions to better put our study in context of recent studies.

3. Line 185: Can you briefly discuss the level of agreement between the omega that you estimate and that which is calculated by the model? I am curious to know how far off they are.
Yes, we had already calculated the correlation coefficients between the model calculated vertical motion and the diagnosed vertical motion at each grid box and pressure level and averaged over latitude bands to ensure our omega equation solver was reliable. We found that between 30–60°N the correlation coefficients were 0.84 at 700 hPa and exceeded 0.9 at 500 hPa. We have added text about this to section 3.2

Related to this, in Section 365, are your discussing the model-produced omega or that which you estimate from this equation?

In section 6.5 we first discuss the model vertical velocity (first two paragraphs), we then compare the model and the calculated omega (third paragraph) before analysing the contributions for the different terms. We have revised the first sentence of section 6.5 (line 365) to make it clear that in this paragraph we discuss the model produced omega.

4. Line 223: From the figure, it is difficult to see what is occurring for low-level baroclinicity (e.g., below 900 hPa), because of the contours. Is it getting stronger or weaker near the surface?
This is a good point and we agree that this is hard to see from the figures. We have added an extra panel to Figure 2 showing the 950-hPa zonal mean temperature in both experiments which shows that the temperature gradient in the mid-latitudes does not change notably with warming. We also revised Figure 3 to hopefully make the low levels clearer and we also discuss the low-level changes in more detail in the text.

5. Line 225: Be careful in how about how you state this. I agree that stratospheric baroclinicity is likely to have a small role. But near the tropopause things are less clear, see for instance: Yuval, J. and Y. Kaspi, 2016: Eddy Activity Sensitivity to Changes in the Vertical Structure of Baroclinicity. J. Atmos. Sci., 73, 1709–1726
Thanks for point this out. We have slightly revised this sentence.

6. Line 270: This latitude shift is going to have a big impact on the precipitation (e.g.,Booth et al.2018).
Yes we agree and this can be seen in Figure 2a which shows the peak in mid-latitude precipitation moves polewards
It seems a bit confusing that the storm track as a whole shifts polewards, but the strongest events initialize closer to the equator. This likely relates to the added role of moisture in driving the strength?
Firstly, we think there may have been some confusion as to what was being compared here as the text was not very clear and as such we have revised this paragraph. Secondly, Figure 1 included in this response can help clarify this point. Figure 1 shows that there are lots of generally weak extra-tropical cyclones that develop poleward of 60°N which result in the mean genesis latitude of all storms being more poleward than when the 200 strongest storms are considered. The fact that the strongest storms form more equatorward is potentially due to more moisture but is also likely due to that fact that these storms develop and track through regions of large Eady growth rate and baroclinicity which storms forming north of 60°N do not do.

Booth, J. F., Naud, C. M., and J. Jeyaratnam, 2018: Extratropical cyclone precipitation lifecycles: A satellite-based analysis. Geophysical Research Letters, 45, 8647-8654.

7. Line 501: Tierney et al. (2018) also documented this shift. My question: can you speculate on what impact this for people, either on the hazards created by the cyclones or the interactions between the cyclone mid-level and upper-level circulation. Can we speculate that in a warmer world the storms mid-level disconnects from the upper-level– which could have a big impact on storm behavior? Isn't this what Tierney et al. and Kirshbaum et al find with their baroclinic wave studies? Or are is this change in the structure of the cyclones in warming simulations just a

curiosity that is of interest to the dynamics community? I think it is the former, but things like this should be stated and discussed a bit so if you want the paper to reach outside the cyclones community.

We now add that Tierney et al also find this shift in the position of the low-level anomaly relative to the upper level anomaly. We want to avoid too much speculation in the paper, however, we think that there is now a reasonable amount of evidence to suggest that in a warmer world the low-level and upper-level PV anomalies of extra-tropical cyclones would interact less / become detached which could change the structure of the storms. This is a topic we plan to investigate more in future work.

**References**

H. F. Dacre, M. K. Hawcroft, M. A. Stringer, and K. I. Hodges. An extratropical cyclone atlas: A tool for illustrating cyclone structure and evolution characteristics. *Bull. Amer. Meteor. Soc.*, 93 (10):1497–1502, 2012.

T. Jung, M. J. Miller, T. N. Palmer, P. Towers, N. Wedi, D. Achuthavarier, J. M. Adams, E. L. Altshuler, B. A. Cash, Kinter I., et al. High-resolution global climate simulations with the ECMWF model in Project Athena: Experimental design, model climate, and seasonal forecast skill. *J. Climate*, 25(9):3155–3172, 2012.

---

## Author Comment (AC2) · 6 Nov 2019

**Response to Reviewers - "The characteristics and structure of extra-tropical cyclones in a warmer climate"**

Victoria A. Sinclair, Mika Rantanen, Päivi Haapanala, Jouni Räisänen and Heikki Järvinen

November 6, 2019

We thank the reviewer for their constructive comments on our submitted manuscript. We have copied the comments of reviewer 2 in black here and include our response to each individual comment in blue.

**Reviewer 2**

This is a very nice study looking at aquaplanet model simulations with a complex model. This provides a step in the model hierarchy between fully complex models (e.g. the CMIP models) and the very idealised models such as a baroclinic channel models. By using a Lagrangian feature tracking algorithm and looking at the lifecycles of extratropical cyclones, the authors have investigated the changing intensity and structure of the 200 most intense cyclones. The paper is well-written and the figures are very clear. I have a few minor comments and suggestions to make.

1. Line 208: The CMIP5 model projections are not shown here so it would be good to refer to a paper that shows these. We have added a reference here and also put the temperature change obtained in our experiments in context of predictions from CMIP5

2. Line 213: Is the poleward change of 2.2 degrees significant? This is much smaller than the shift in the jet. The maximum in the zonal mean precipitation moves 2.2 degrees polewards whereas the eddy driven jet (taken as the zonal mean zonal wind speed at 700 hPa) moves from 37.6N to 40.9N, a poleward shift of 3.3 degrees, therefore, we do not think the jet shift is much smaller than the shift in the maximum precipitation, especially given the model resolution this equates to one more grid point.

3. Line 219: Is there an explanation for the much lower average MSLP? Is this an issue with OpenIFS or something to do with the aquaplanet set up? This is an artefact of how we created the initial conditions for the aqua-planet which we explain in sections 2.2. We have revised this sentence to clarify this issue.

4. Lines 317-321: It would be good to see figures for the changes in the wind speeds. Both here and in the precipitation section, I think a useful addition would be some analysis of the footprints of the most intense winds and precipitation and how this changes. These footprints were considered in the Tierney et al 2018 and Pfahl et al 2015 papers. We now include a figure showing the

900-hPa wind speeds in CNTL and the response to warming which we previously discussed in section 6.2. The text about wind speeds has been expanded and moved to its own subsection.

5. Line 350: I think this sentence should be reworded slightly – it seems that what is consistent is that the condensation from the precipitation gives more latent heating and stronger PV anomaly (rather than the latent heating leading to more precipitation). We have revised this sentence.

6. Line 357: It would be good to make it clearer here and elsewhere in the paragraph when it is referring to changes in the SST4 experiment. Good point. We have revised this section to be clearer.

7. Line 376-377: I think it might be good to say the warm conveyor belts are further poleward relative to the propagation direction rather than the cyclone centre since the cyclones have been rotated for compositing. The rotation means that all cyclones in the composites are propagating due east which means that in the composite mean image the propagation vector is at the same point (in the meridional direction) as the cyclone centre. As we are discussing the rotated composite mean we have not changed this.

8. Line 416: I think the wrong figure panel is referenced here – it should be 10d.
Thank you, this was a mistake and we have now corrected it.

9. Section 7: It would be nice to see greater discussion of this study in the context of previous literature. For example, how do these results compare with, e.g. Pfahl etal 2015, Tierney et al 2018. Are there any other papers that analyse the structure of extratropical cyclones in the future? Two that I can think of are Yettella and Kay 2017 and Michaelis et al 2017. We have now expanded the conclusions section to include more discussion and comparison to previous studies and reference both of these suggested papers.

10. Section 7: Also, are there any caveats with the study – would the results change if you looked at the 500 most intense storms or the medium intensity storms? There are certainly caveats associated with this study, which we have added discussion about to the conclusions. We also now include the results of how the median cyclones change with warming in supplementary material.

References:
Yettella V, Kay JE. How will precipitation change in extratropical cyclonesas the planet warms? Insights from a large initial condition climate model ensemble.Clim Dyn. 2017;49(5–6):1765–81.

Michaelis AC, Willison J, Lackmann GM, Robinson WA. Changes in Winter North Atlantic Extratropical Cyclones in High-Resolution Regional Pseudo–Global Warming Simulations. J Clim. 2017 Jun 6;30(17):6905–25.

---

## Author Comment (AC3) · 6 Nov 2019

**Response to Reviewers - "The characteristics and structure of extra-tropical cyclones in a warmer climate"**

Victoria A. Sinclair, Mika Rantanen, Päivi Haapanala, Jouni Räisänen and Heikki Järvinen

November 6, 2019

We thank the editor for their comments on our submitted manuscript. We have copied the comments in black here and include our response to each individual comment in blue.

**Editor's minor comment**

1. Figure 6 shows the difference between SST4 and CNTL for the 900–700 hPa layer mean potential vorticity. There are two maxima, for example in Fig.6c (t=0), the first near the occlusion point (in the northeast sector of the cyclone) and the second near the bent-back front (close to the composite center, corresponding to the vorticity maximum identified by the tracking). However, the corresponding equivalent for precipitation, Fig. 8g (t=0), shows only one maxima, which is in the north-eastern sector of the cyclone. This suggests, that the potential vorticity seen in Fig.6c in the north-eastern sector of the cyclone is formed by enhanced diabatic processes, while the second potential vorticity anomaly near the bent-back front is resulting from enhanced advection. This difference could be explained by the linkage between the cold and the warm conveyor belts. The positive potential vorticity anomaly, which is diabatically generated in the air below the rising warm conveyor belt, is advected by the cold conveyor along the bent-back front, where it contributes to the enhanced potential vorticity gradients (corresponding to higher wind speed near the tail of the bent-back front of the cyclone). When the linkage is accelerated, the first low-level potential vorticity maximum is explained by enhanced diabatic processes and the second by accelerated advection by the cold conveyor belt, resulting in only one maxima in the precipitation pattern but two in the potential vorticity pattern, which is in agreement with the here presented composites (Fig.6 and Fig.8). The linkage between the conveyor belts has been described in an idealized setting in Schemm and Wernli (2014) and is summarised in their Figure 9.

   Schemm, S. and H. Wernli, 2014: The Linkage between the Warm and the Cold Conveyor Belts in an Idealized Extratropical Cyclone. J. Atmos. Sci., 71, 1443–1459, `https://doi.org/10.1175/JAS-D-13-0177.1` (A video that helps to illuminate the linkage is provided at `https://journals.ametsoc.org/doi/suppl/10.1175/JAS-D-13-0177.1`)

Thank you for sharing this insight with us. This does appear to be a plausible explanation but without performing Lagrangian diagnostics we cannot prove that this is certainly the case. However, we have added a comment about this potential mechanism to section 6.5 of the revised manuscript.

2. A couple of suggested additional literature that seems to be in agreement with the findings of the submitted manuscript. Regarding the changes eddy intensity:

Paul A. O'Gorman 2010: Understanding the varied response of the extratropical storm tracks to climate change. Proceedings of the National Academy of Sciences Nov 2010, 107 (45) 19176-19180; DOI: 10.1073/pnas.1011547107

O'Gorman, P.A. and T. Schneider, 2008: Energy of Midlatitude Transient Eddies in Idealized Simulations of Changed Climates. J. Climate, 21, 5797–5806, `https://doi.org/10.1175/2008JCLI2099.1`

And in agreement with the fact that the large-scale eddies appear to stabilize the tropopshere in a warmer climate:

Korty, R.L. and T. Schneider, 2007: A Climatology of the Tropospheric Thermal Stratification Using Saturation Potential Vorticity. J. Climate, 20, 5977–5991, `https://doi.org/10.1175/2007JCLI1788.1`.

Thanks for these suggestions. We did not include these as these are studies considering the whole atmosphere in an Eulerian framework, rather than in a cyclone relative framework which makes direct comparisons difficult. However, based on the reviewers comments we have added an extra section on the asymmetry of vertical motion to help link to earlier work and we now also compared our results to a wider range of relevant studies in the conclusions.

---

## Author Response (AR2)

**The characteristics and structure of extra-tropical cyclones in a warmer climate**
**By Victoria Sinclair, Mika Rantanen, Päivi Haapanala, Jouni Räisänen and Heikki Järvinen**

**Response to Reviewer 1.**

Thank you for pointing out these small typos in our manuscript. We have revised the manuscript accordingly (see below – our response in blue). We have also included a track changed version of the manuscript.

- Line 248: You write: "… and the average surface pressure of 985.4 hPa results as it is the average pressure at the actual surface height in the …" I think there should be a comma after results. We have added the comma here.

- Line 260: You write:  "With the mid-troposphere, the Eady growth decreases slightly with warming, …" The word rate needs to be added after the word growth. Or jut replace Eady growth rate with its symbol: sigma. We have added "rate".

- Line 646: No need to spell out RCP here. We now just use the abbreviation here.

- Line647: You write:  "The relative increase is thus smaller than found in our aqua-planet simulations " Do you need to add the word "that" after the word "than"? Yes you are correct, We have added "that".

[revised manuscript text omitted]